# A novel cloud removal method by fusing features from SAR and neighboring optical remote sensing images

Yuyao Wang[1,2,3], Jiehai Cheng[1,2]*

1 School of Surveying & Land Information Engineering, Henan Polytechnic University, Jiaozuo, China, 2 Henan Province Spatial Big Data Acquisition Equipment Development and Application Engineering Technology Research Center, Henan Polytechnic University, Jiaozuo, China, 3 Earthquake Administration of Bayan Nur, Bayan Nur, China

* chengjiehai@hpu.edu.cn

## Abstract

Optical remote sensing images were prone to extensive cloud coverage, especially under mountainous conditions with frequent weather changes. To address this issue, this paper proposed a novel cloud removal method that integrated features from SAR and neighboring optical remote sensing images. The method was based on a deep CGAN network, leveraging both local and global features of SAR images as well as edge features of optical remote sensing images to perform coarse cloud removal. Building upon the coarse cloud removal, the spectral features of neighboring optical remote sensing images were utilized for refined cloud removal. The experimental results showed that the proposed coarse-to-fine cloud removal method achieved a satisfactory cloud removal performance for optical remote sensing images. The RMSE, SAM, mSSI, and CC values were 0.0391, 0.0729, 0.9221, and 0.9537, respectively. Compared with other classical methods, each of these four metrics improved by at least 0.0119, 0.0438, 0.0217, and 0.0240, respectively. Moreover, the proposed method demonstrated superior performance in terms of boundary smoothness, cloud shadow elimination, spectral consistency, and spatial detail restoration for cloud removal in images with different underlying surface conditions. The effectiveness of the proposed method will improve the quality of cloud removal in optical remote sensing images.

## 1 Introduction

Since 1972, optical remote sensing (RS) images, represented by Landsat satellite data, have been widely used in the industry [1]. However, the imaging system used to acquire optical RS images operates in a passive mode, which renders it highly susceptible to weather conditions. As a result, optical RS images frequently exhibit extensive cloud cover [2,3]. In southern China during the summer season or under

**Data availability statement:** All relevant data are available from the figshare database (URL: https://figshare.com/s/a0a94948992299d8f421).

**Funding:** This work was supported in part by the National Natural Science Foundation of China under Grant 42171299; in part by the Surveying and Mapping Science and Technology 'Double First- Class' Discipline Creation Project under Grants GCCYJ202409 and BZXCG202403; and in part by the Natural Science Foundation of Henan Province under Grant 162300410122.

mountainous terrain conditions, such cloud cover in optical RS images is particularly pronounced, which severely limits its applications in crop mapping, land use/land cover change monitoring, and natural disaster information extraction.

Experts and scholars conduct extensive research on the issue of cloud removal in optical RS images, and many solutions are proposed. In the early stages, cloud removal is primarily achieved based on the cloud-covered optical RS images themselves. Most of the initial cloud removal methods are model-driven. For example, Liu et al. utilize a cloud physics model for cloud removal, while Zi et al. combine imaging models with deep learning methods to achieve cloud removal [4,5]. It is proven that such methods are only effective for thin cloud removal [6–8]. When the cloud layer reaches a certain thickness, the optical RS images lose excessive information, which renders model-driven cloud removal methods ineffective. Subsequently, researchers explore data-driven approaches for cloud removal. Due to the lack of auxiliary data, these methods rely on the local neighboring information within the cloud-covered optical RS images themselves. For example, the nearest neighbor interpolation method proposed by Siravenha et al., and the neighborhood similar pixel interpolation methods proposed by Zhu et al. and Chen et al. are proposed to perform well in regions with small-scale cloud cover [9–11]. These methods perform well in regions with small-scale cloud cover. However, due to the limited available information in a single optical RS image, precise reconstruction cannot achieve through simple interpolation in practical applications, thus limiting their applicability. In cases where cloud cover in RS images is minimal and scattered, and the underlying features are relatively simple, cloud removal methods based solely on the cloud-covered optical images can reconstruct missing information without the need for any reference images [12]. However, these methods often perform poorly in areas with extensive cloud cover or complex textures, as they do not utilize information from the cloud-covered regions of the RS images themselves.

Zheng et al. and Mao et al. proposed using RS images from other sensors to supplement information in cloud-covered areas, thereby achieving cloud removal effects [13,14]. Initially, different optical sensor images are used to assist in cloud removal. Wang et al. utilize Landsat multi-spectral RS images to remove thick clouds from hyper-spectral RS images, demonstrating the potential of using optical sensor images of different types to assist in cloud removal [15]. Zhang et al. and Zhao et al. utilize MODIS optical RS images to perform cloud removal processing on Landsat and Sentinel-2 optical RS images, respectively [16,17].MODIS optical RS images have an extremely high temporal resolution, making it easier to find cloud-free optical RS images of the area, thereby addressing the issue of obtaining cloud-free optical images within adjacent time periods [18]. This method is simple and straightforward, but its application is limited by the following factors: (i) the two optical RS images involved in the cloud removal operation may differ in spatial resolution, spectral characteristics, and imaging time; (ii) cloud-free optical RS images under complex terrain conditions are relatively rare. Since SAR images are not affected by weather conditions, using SAR images to supplement the information of cloud-covered areas in optical RS images is a common practice for cloud removal in optical RS [19].

For example, Gao et al. fuse simulated optical RS images with SAR data for cloud removal. The experimental results demonstrate that SAR not only provides reference information for the missing parts of optical RS images but also allows for the control of the global structure during the reconstruction process [20]. Xu et al. utilize SAR images to reconstruct cloud-covered areas in optical RS images by combining global and local approaches. This study demonstrates through extensive experiments that SAR images have a strong capability to provide information about the underlying features at various levels of cloud cover, making the use of SAR images to assist in cloud removal from optical RS images a practical solution [21]. Compared with traditional methods that are only applicable to thin clouds, SAR, with its microwave penetration capability, can not only handle thin cloud obstruction but also provide structural information support under thick cloud coverage (such as cumulus and stratocumulus), where optical imagery suffers from severe loss of ground information, thereby effectively assisting in cloud removal for heavily cloud-covered areas [22]. It is well known that SAR images and optical RS images exhibit significant differences in data characteristics due to their different imaging modalities [23]. Relying solely on SAR images for cloud removal in optical RS images can result in significant distortions in spatial details and spectral characteristics [24].

Regardless of how complex the terrain conditions of a region are, or how high the temporal resolution requirements for the studied objects in optical RS images may be, it is always possible to find cloud-free optical RS images for areas that are cloud-covered [25]. These cloud-free images may differ in imaging time from the optical RS images that require cloud removal and may also come from nearby temporal optical RS images from other sensors. For example, in the context of mapping rice using optical RS images, it is generally possible to obtain cloud-free optical RS images corresponding to cloud-covered areas throughout the entire phenological period of rice growth. For convenience, these cloud-free optical RS images are referred to as neighboring optical RS images. Compared to SAR images, the features of neighboring optical RS images are more similar to those of the cloud-covered optical RS images [26]. By using SAR images for cloud removal in optical RS images and then integrating neighboring optical RS images, the cloud removal effectiveness is greatly enhanced. However, existing methods have yet to effectively integrate SAR image features with neighboring optical image features for cloud removal in optical RS images, thus limiting the enhancement of cloud removal performance. To tackle this issue, this study proposes a novel cloud removal method that integrates SAR and neighboring optical RS image features. The main contributions of this study are as follows:

(1) Developing a coarse-to-fine cloud removal method to enhance the quality of cloud removal in optical RS images.

(2) Demonstrating the effectiveness of integrating multidimensional features from SAR and optical RS images in enhancing cloud removal performance.

(3) Exploring the applicability of CGAN for cloud removal in optical RS images.

The remainder of this study is organized as follows. Section 2 explains the principles of the proposed method. Experimental results and analysis are presented in Section 3. Section 4 discusses the experimental results. Finally, Section 5 concludes this paper.

## 2 Methodology

### 2.1 Overview of the proposed framework

The proposed framework (as illustrated in Fig 1) consists of two stages: (i) Coarse cloud removal from optical RS images. A deep learning model is employed to automatically detect cloud-covered regions in the optical RS image. The SAR image is registered with the cloud-covered optical image to generate a series of $n \times n$ image patch pairs for both cloudy and cloud-free regions. Simultaneously, the SAR image is input into a Vision Transformer (ViT) model to obtain the Global Feature Map. The SAR image and the cloud-covered optical image are further processed through the Edge Information Generation Model to produce an Edge Image. The image patch pairs, Global Feature Map, and Edge Image are then fed

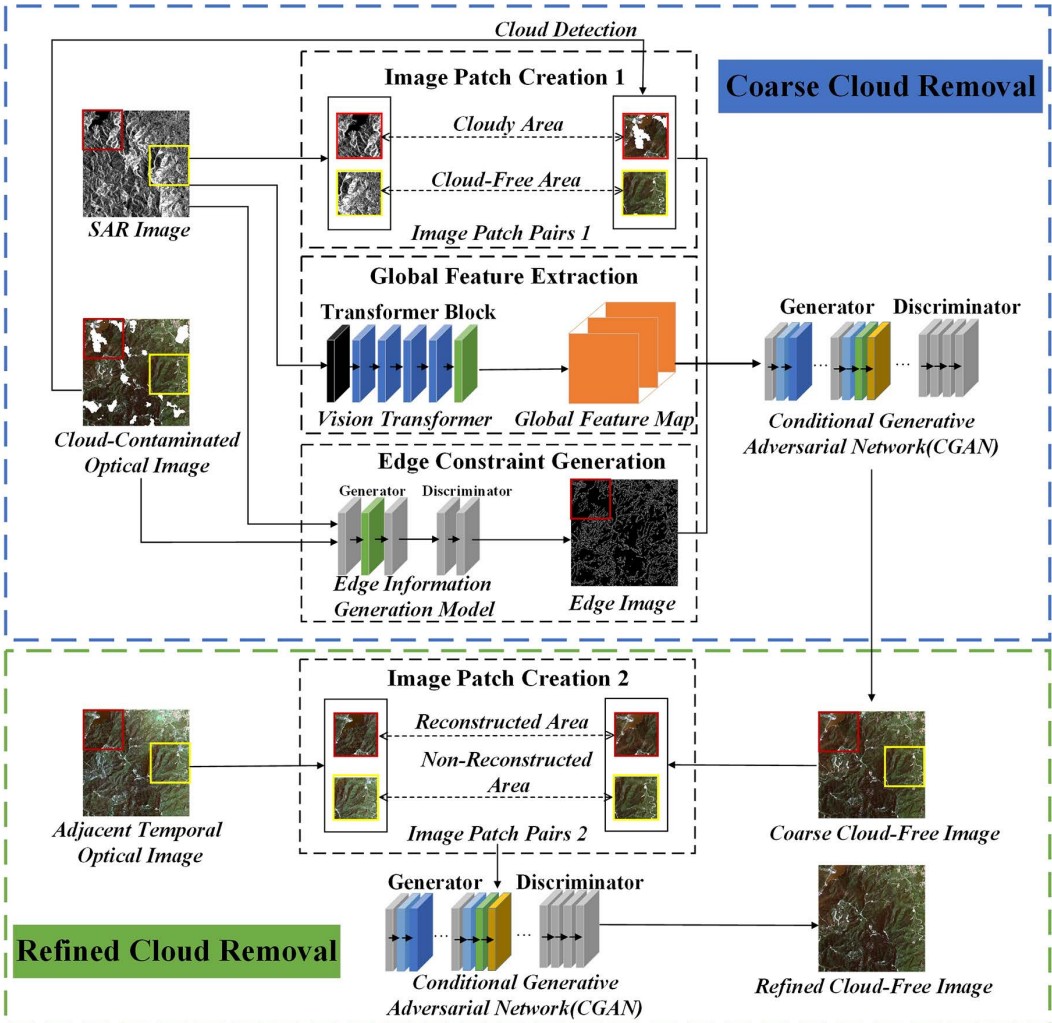

**Fig 1. Flowchart of the proposed model for cloud removal.**

into a Conditional Generative Adversarial Network (CGAN) to generate the coarse cloud-free image. (ii) Refined cloud removal for optical RS images. Neighboring optical images of the cloud-covered optical image are collected and aligned with the coarse cloud-free image. Image patch pairs are generated based on the reconstructed and non-reconstructed regions between the coarse cloud-free image and the neighboring optical images. These pairs are input into the CGAN model to produce the refined cloud-free image.

## 2.2 The coarse cloud removal of optical remote sensing images

**2.2.1 Image patch creation 1.** The purpose of establishing image patch pairs was to leverage the local feature mapping relationship between SAR images and optical RS images to restore cloud-covered regions in optical RS images. To achieve this, cloud-covered regions in the optical images needed to be detected first. A series of 128 × 128 image patch pairs were randomly generated from cloud-covered and cloud-free areas, with all generated image patch pairs belonging to the same temporal image. The two types of regions were respectively split into training and validation sets at a ratio of

7:3. The cloud-free image patch pairs were input into a deep neural network to learn the local feature mapping relationship between SAR and optical RS images. The learned mapping was then applied to the cloudy image patch pairs to restore the cloud-covered areas within the cloudy image patches.

Cloud detection in optical RS images was performed using the Refined U-Net model [27]. The model was trained on a synthetic cloud dataset and the GF1-WHU remote sensing cloud dataset [28], which includes 120 scenes covering various global land cover types and multiple cloud conditions. These conditions include low-level stratocumulus, mid-level altocumulus, high-level cirrus clouds, as well as cumuliform clouds with significant vertical development such as cumulus and cumulonimbus clouds, featuring distinct layering and varying thickness characteristics. The cloud masks in the dataset included two categories: "cloud" and "cloud shadow." During training, the dataset images were cropped into 128×128 patches to match the model input size. To prevent model overfitting, various data augmentation techniques were applied to expand the training data, including random horizontal flips, vertical flips, brightness adjustments, and slight Gaussian noise perturbations. After training, the model was used to predict cloud-covered images, resulting in cloud masks containing both clouds and cloud shadows.

**2.2.2 Global feature extraction.** Image patch pairs1 focused on the local features of the image but neglected the global features. SAR images provided more comprehensive global features, which not only captured the overall structure and semantic information of the image but also offered a holistic understanding of the entire image content, providing reliable support for cloud removal tasks. In this study, the Vision Transformer (ViT) model was used for global feature extraction [29].

The ViT model architecture is shown in Fig 2 and can be divided into three parts.

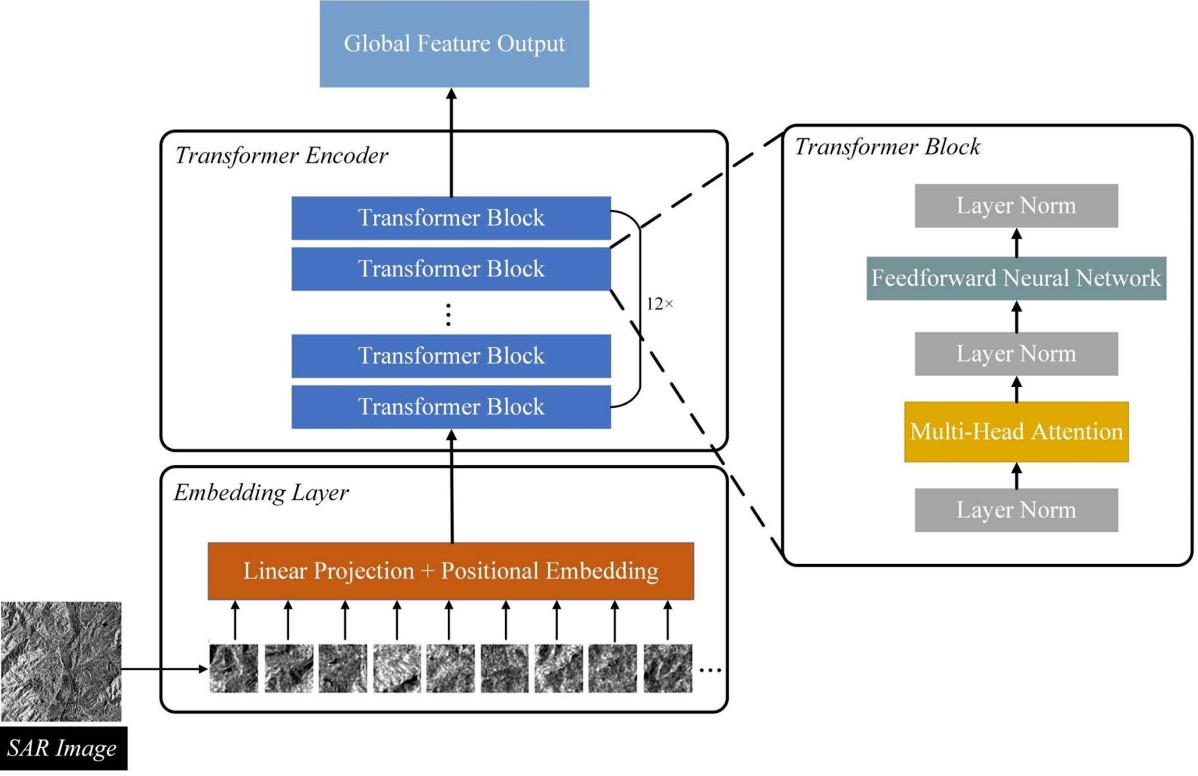

**Fig 2. Network structure of Vision Transformer (ViT).**

(1) Linear Projection of Image Patches (Embedding Layer)

To transform the image data into a format suitable for input into the Transformer model, the original image was divided into several 64×64 image patches. Each patch was flattened, converting it from a two-dimensional matrix to a one-dimensional vector of fixed length. These vectors underwent a dimensional transformation via a linear transformation layer to match the input requirements of the Transformer encoder. To enable the model to capture the spatial position information of the image patches, fixed positional encodings were added to the embedding vectors after the linear transformation. Ultimately, the representation of each image patch (Embedded Patches) included the transformed feature vector and the corresponding positional information.

(2) Transformer Encoder

The Embedded Patches were input into the Transformer encoder, which consisted of 12 identical Transformer Blocks. Each Transformer Block mainly included a Layer Norm, Multi-Head Attention, and a position-based Feedforward Neural Network. Among these, the Multi-Head Attention mechanism was key to the Transformer's strong global modeling ability, allowing the model to process various information from different subspaces.

(3) Feature Output Layer

After processing by the Transformer encoder, the feature representations of the image patches were integrated into global features. A global average pooling operation was applied to compute the average of all image patch features, generating a fixed-dimensional vector that served as the global feature representation of the entire image.

Currently, the ViT model, as a novel image processing approach, introduces the Transformer architecture to extract long-range dependencies within image sequences through a multi-head attention mechanism, fully exploiting contextual information between images and providing a new solution for global feature extraction. Compared with traditional convolutional neural networks (CNNs), ViT offers greater advantages in processing SAR data. CNNs are limited by their local receptive fields and struggle to effectively model long-distance dependencies in images, whereas SAR images typically contain complex structures and spatial correlation features that require global modeling capabilities [30]. ViT captures relationships between any positions in the image through self-attention mechanisms, making it better suited for extracting the global structural features of SAR data. Therefore, this study utilizes ViT to extract global features, which better facilitates improving the overall reconstruction performance in cloud removal tasks.

**2.2.3 Edge constraint generation.** The imaging methods of SAR and optical RS images were different, leading to significant discrepancies in the representation of the same objects on the two types of images. Fig 3 showed the differences in mountainous terrain between SAR and optical RS images. There were notable differences in the edge information between the two. These differences in edge information could have introduced some uncertainty in the cloud removal process [31]. To address this, an Edge Information Generation Network [24] was employed in this study to automatically generate edge maps for the optical RS images to be processed. These edge maps were used as features to constrain the edge information differences between SAR and optical RS images.

The structure of the Edge Information Generation Network is shown in Fig 4, consisting of two parts: the generation network and the discrimination network. SAR images, simulated cloud-covered optical RS images, simulated cloud-covered optical RS image edge maps, and cloud masks were input into the generation network to produce edge maps. The generated edge map was then input, along with the true edge map, into the discrimination network, which optimized the parameters of the generation network by distinguishing between real and generated edge maps, allowing the generation network to produce results that were closer to the true edge map. The true edge map was generated using the Canny edge detection algorithm. This structure was widely used in image interpretation tasks and had been proven to be highly effective in handling similar image generation tasks [24]. After training, SAR images, real cloud-covered images, real cloud-covered image edge maps, and cloud masks were input for prediction to obtain edge maps for real cloud-covered scenarios.

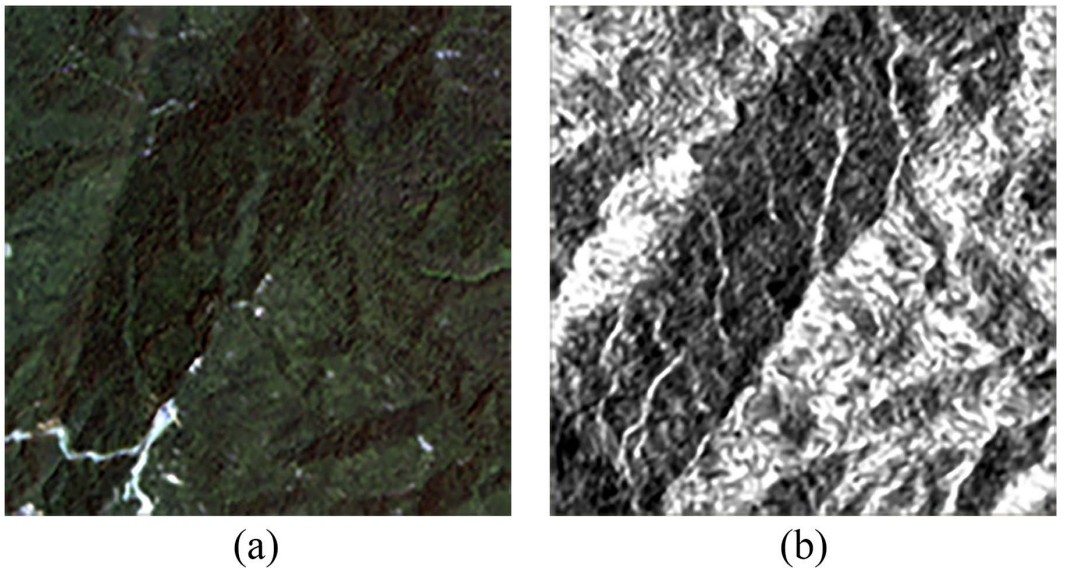

(a)                                          (b)

**Fig 3. Examples of Edge Differences in Images.** (a) Optical RS Image (b) SAR Image.

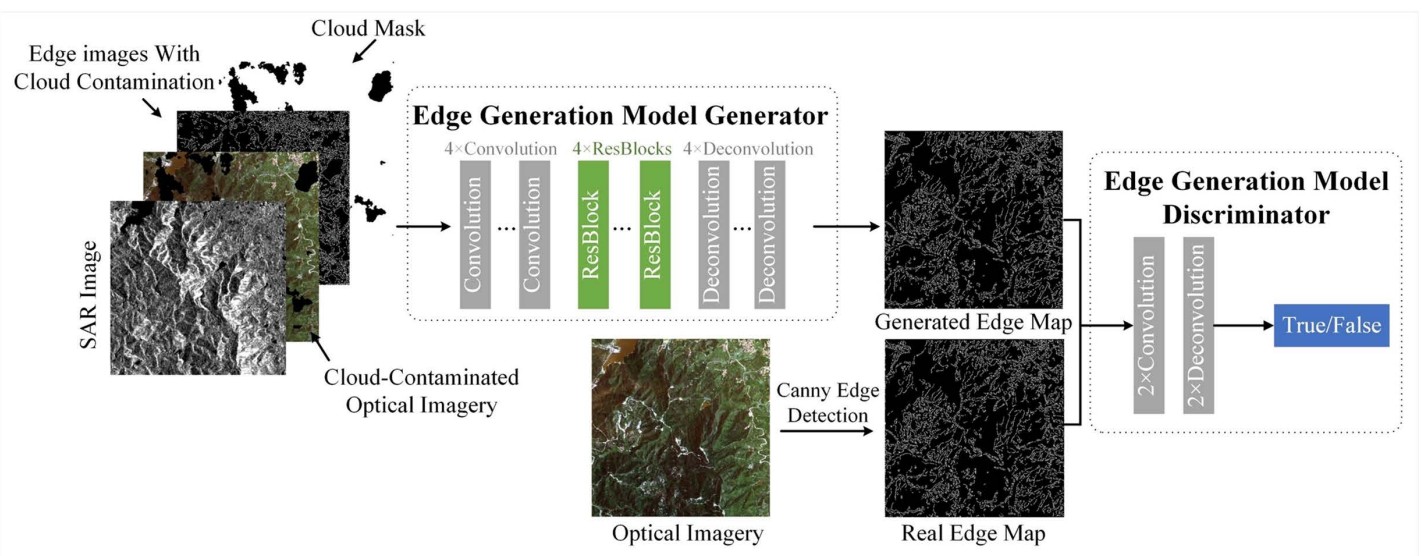

**Fig 4. Network structure of Edge Information Generation Model.**

## 2.3 Refined cloud removal for optical remote sensing images

The spectral features of the neighboring optical RS image were used to refine the cloud removal process on the coarse cloud-free image. The neighboring optical RS image had the same coverage as the optical RS image to be processed, was temporally adjacent, and was free of cloud cover; it may have come from different sensors. As shown in Fig 1, the purpose of establishing image patch pairs 2 was to use the local spectral feature mapping relationship between the neighboring optical RS image and the coarse cloud-free image to perform refined cloud removal on the coarse cloud-free

image. Using the cloud mask, the image was divided into reconstruction and non-reconstruction regions. The reconstruction region refers to areas labeled as "cloud" in the cloud mask, which require further refinement in the coarse cloud-free image. The non-reconstruction region refers to areas labeled as "non-cloud" in the cloud mask, which do not require refinement in the coarse cloud-free image. Based on this, a series of 128 × 128 image patch pairs were generated. The image patch pairs from the non-reconstruction region were input into a deep neural network to learn the local spectral feature mapping relationship between neighboring optical RS images and the coarse cloud-free image through supervised learning. During training, the non-cloud regions of the coarse cloud-free image were used as supervisory labels, and the L1 loss function was employed to constrain the spectral difference between the model output and the true image. Subsequently, the learned mapping was applied to the image patch pairs in the reconstruction region to perform refined cloud removal on the coarse cloud-free image, ultimately obtaining the refined cloud-free image.

## 2.4 CGAN

This study used Conditional Generative Adversarial Networks (CGAN) for cloud removal in both the coarse and refined stages, with slight differences in its application for model optimization. CGAN, as a type of generative adversarial network, efficiently integrated multi-source features and had been proven to perform excellently in learning complex features, making the generated cloud-free images more accurate [32].

**2.4.1 Architecture.** The CGAN consisted of two parts: the generator network and the discriminator network. Feature information was input into the generator network, which output generated results that the discriminator network could not distinguish from real ground truth images. The discriminator network tried to distinguish the generated results from real ground truth images, pushing the generator network to improve. The two networks alternated optimization, forming adversarial learning. Finally, when the two networks reached a balance, the generator network was used to complete the reconstruction task.

The structures of the generator and discriminator networks were shown in Fig 5, and their architectures were discussed in Section 4.2. The generator network adopts a 14-layer U-Net structure with modifications: all batch normalization (BN) layers were removed. BN layers can introduce unstable statistical biases during small-batch training and alter the original distribution of feature maps. This may cause structural distortion or artifacts in image reconstruction tasks that are sensitive to global information such as brightness and contrast, thereby affecting the final reconstruction quality. To better preserve the original global feature distribution of the images, BN layers were removed in the network design. The discriminator was a simple four-layer convolutional neural network. Real images and reconstruction results were fed into the discriminator, which independently determined whether each input was "real" or "fake."

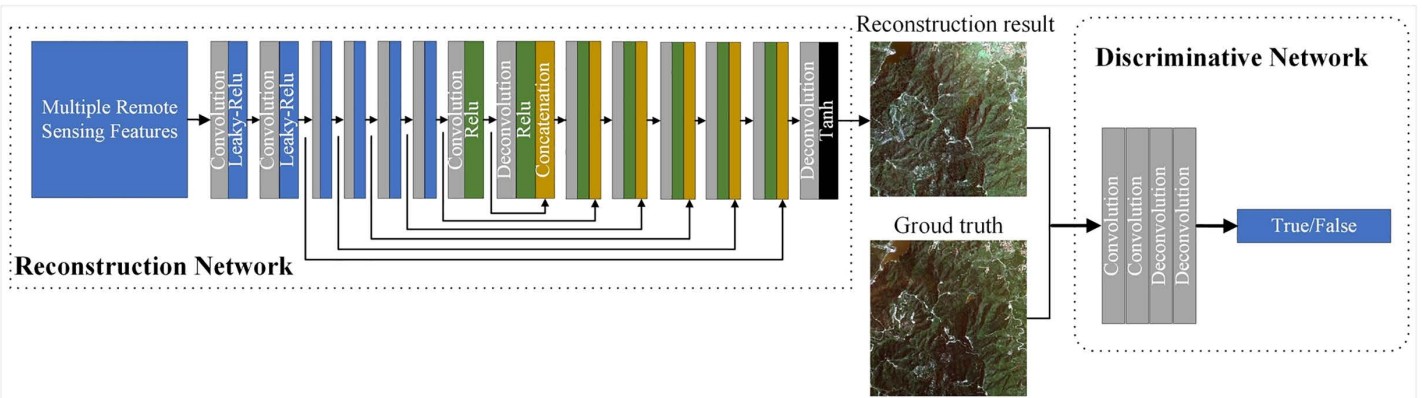

**Fig 5. Network structure of CGAN.**

**2.4.2 Optimization.** To improve the quality of cloud-removed images, the original CGAN was optimized, focusing on the loss function. In the coarse cloud removal stage, the traditional L1 loss function was used to constrain the initial cloud removal results. For the refined cloud removal stage, a composite loss function was designed to optimize the final generated results, combining adversarial loss, L1 loss, perceptual loss, edge loss, and local loss. This ensured that the generated image not only appeared visually realistic but also closely resembled the real image in terms of structure and detail.

(1) Adversarial Loss

Adversarial loss ensures that the images generated by the generator are as indistinguishable as possible from real images by the discriminator network. For the generator network, this study uses a binary cross-entropy-based adversarial loss function:

$$L_{CGAN} = -E_{z \sim p_z(z)} \left[ logD\left(G(z)\right) \right] \tag{1}$$

Among them, $G$ is the generator, $D$ is the discriminator, and $z$ is the random noise.

(2) L1 Loss

L1 loss is used to measure the pixel-wise differences between the generated image and the real image, enhancing the overall structural consistency of the generated image and controlling the global structure of the network. It is defined as follows:

$$L_{L1} = E_{x \sim p_{data}(x), z \sim p_z(z)} \left[ \left| x - G(z) \right|_1 \right] \tag{2}$$

Where $x$ is the real image.

(3) Perceptual Loss

Perceptual loss calculates the differences between the generated image and the real image using high-level feature maps from a pre-trained convolutional neural network, VGG16, thereby preserving more detailed visual features. The specific calculation formula is as follows:

$$L_{L1} = E_{x \sim p_{data}(x), z \sim p_z(z)} \left[ \left| x - G(z) \right|_1 \right] \tag{3}$$

Where $\phi_i$ denotes the feature representation of the i-th layer of the pre-trained VGG16 model.

(4) Edge Loss

Edge loss is used to constrain the edge structure of the generated image. It calculates the edge map of the image using the Canny algorithm and then measures the differences between the edge maps. It is defined as follows:

$$L_{edge} = E_{x \sim p_{data}(x), z \sim p_z(z)} \left[ \left| canny(x) - canny(G(z)) \right|_1 \right] \tag{4}$$

Where $canny()$ is the Canny algorithm used to compute the image edges.

(5) Local Loss

The loss functions defined above primarily focus on reconstructing the cloud-covered image from a global perspective. Additionally, a loss function should be designed to specifically emphasize the reconstruction quality in the local

regions affected by clouds. Therefore, this study constructs a local loss function using the cloud mask M for localized reconstruction:

$$L_{local} = \|M \cdot F - M \cdot G\|_1 \#$$ (5)

Where $M$ is the cloud mask, $F$ is the real image, and $G$ is the generated image.

The final composite loss function is defined as the weighted sum of the five aforementioned losses:

$$L_{total} = \lambda_{CGAN} L_{CAN} + \lambda_{L1} L_{L1} + \lambda_{perceptual} L_{perceptual} + \lambda_{edge} L_{edge} + \lambda_{local} L_{local} \#$$ (6)

Where $\lambda_{CGAN}$, $\lambda_{L1}$, $\lambda_{perceptual}$, $\lambda_{edge}$ and $\lambda_{local}$ are the corresponding loss weight coefficients that control the contribution of each loss to the total loss. In the experiments, we set $\lambda_{CGAN}$ =0.0001, $\lambda_{L1}$ =10, $\lambda_{perceptual}$ =1, $\lambda_{edge}$=1 and $\lambda_{local}$ =10. This study achieved a balance among the different loss weights, allowing the generator to produce realistic images while maintaining the structural integrity and detail of the images.

(6) Discriminator Loss Function

The loss function of the discriminator measures its ability to distinguish between real and generated images. It typically includes two components: (i) the loss for real images, which assesses the penalty for classifying real images as fake; (ii) the loss for generated images, which evaluates the penalty for classifying generated images as real.

The total loss of the discriminator is the sum of these two losses:

$$L_D = -E_{x \sim p_{data}}(x)\left[logD(x)\right] - E_{z \sim p_z(z)}\left[\log\left(1 - D\left(G(z)\right)\right)\right]$$ (7)

where $D$ is the discriminator, $G$ is the generator, $x$ is the real image, and $z$ is the random noise.

## 3 Experiments and results

### 3.1 Datasets

The study area was located in the Dabie Mountains of China (Fig 6). The Dabie Mountains were situated in the central-eastern region of China, characterized by complex mountainous terrain and a humid, variable climate. Fog and cloud cover were common phenomena, especially in summer. The primary crop cultivated in the Dabie Mountains was rice. To dynamically monitor rice cultivation in this area using optical RS images, it was essential to address the cloud removal issue.

The data used in the experiments came from two areas, Area A and Area B, within the study region, both belonging to Shangcheng County, Xinyang City. Area A was used for simulated experiments; its optical imagery is of high quality with no significant cloud interference, facilitating the construction of high-quality simulated scenarios and controlled variable quantitative evaluation. Area B contains real cloud coverage conditions, suitable for validating the practical performance of the method, thus complementing Area A. Both areas are dominated by farmland and forest land, featuring typical mountainous agricultural landscapes representative of the RS cloud removal application scenarios focused on in this study. Area A data include GF-2 optical RS images (Fig 6(a)), GF-2 neighboring optical RS images (Fig 6(b)), and GF-3 SAR images (Fig 6(c)), with image dimensions of 3742 × 2947 pixels. The GF-2 optical images were acquired on August 4, 2023, and the neighboring GF-2 optical images on August 19, 2023; both have a spatial resolution of 1 meter. The GF-3 SAR images were acquired on July 25, 2023, with a resolution of 5 meters. Smooth simulated cloud maps covering 30% of the total image area were generated on the optical images using the Simplex noise function. Area B data include GF-2 optical RS images (Fig 6(d)), GF-2 neighboring optical RS images (Fig 6(e)), and GF-3 SAR images (Fig 6(f)), with image dimensions of 2757 × 2671 pixels. The GF-2 optical images were acquired on July 16, 2023, and the neighboring GF-2 optical images on August 8, 2023; both have a spatial resolution of 1 meter. The GF-3 SAR images were acquired on

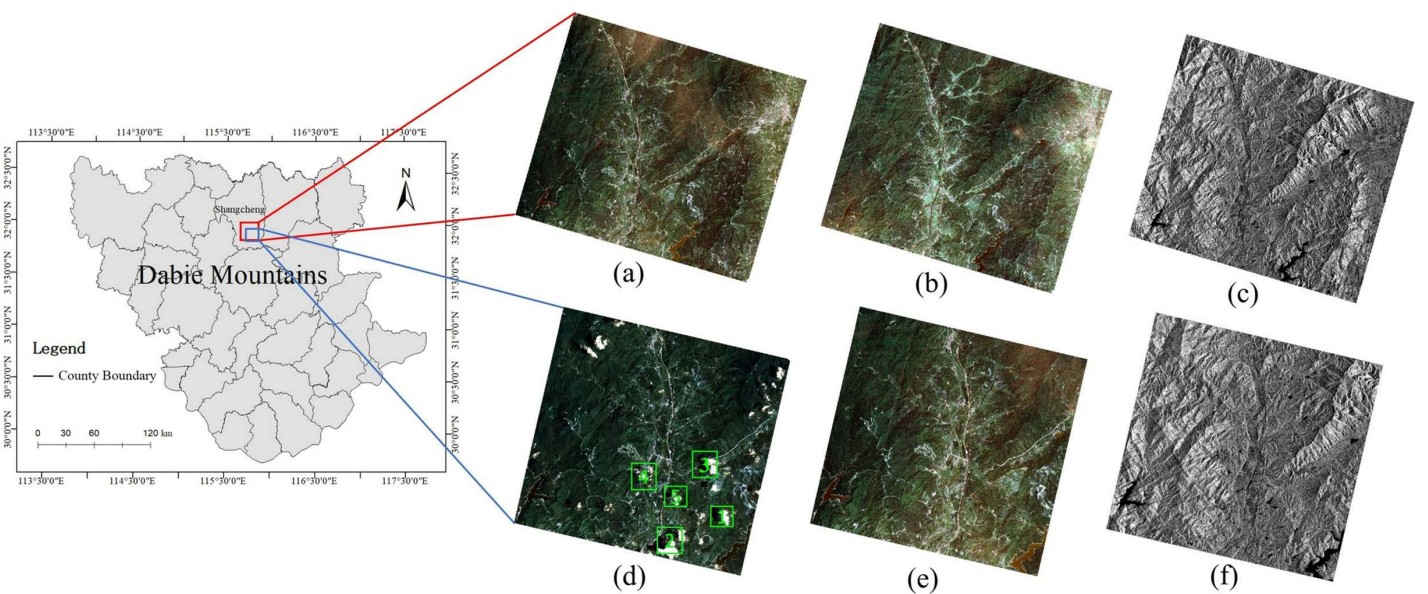

**Fig 6. Data from the Study Area.** (a) Optical RS Image of Area A; (b) Neighboring Optical RS Image of Area A; (c) SAR Image of Area A; (d) Optical RS Image of Area B; (e) Neighboring Optical RS Image of Area B; (f) SAR Image of Area B.

July 4, 2023, with a resolution of 5 meters. All images underwent radiometric calibration, atmospheric correction, geometric correction, and spatial registration, with registration errors controlled within 5 pixels. Nearest neighbor interpolation was applied to unify spatial resolutions. Prior to use, SAR images were filtered using the Lee filter to suppress speckle noise and combined from HH and HV polarization channels to enhance feature representation. Regarding data sources, the GF-2 and GF-3 RS images used in this study were legally purchased through research projects, have legitimate usage licenses, can be used for scientific publication without restrictions, and possess good reproducibility.

### 3.2 Evaluation of cloud removal performance

To quantitatively assess the performance of the generative model, this study selected four metrics: Root Mean Squared Error (RMSE), Spectral Angle Mapper (SAM) [33], Mean Structural Similarity Index (mSSI) [34], and Correlation Coefficient (CC) to evaluate the effectiveness of cloud removal.

(1) Root Mean Squared Error (RMSE)

RMSE measures the difference between predicted values and actual values, with smaller values indicating better model performance. Its calculation formula is as follows:

$$RMSE = \sqrt{\frac{1}{n}\sum_{i=1}^{n}(P_i - T_i)^2}$$

(8)

where $P_i$ is the predicted value, $T_i$ is the true value, and n is the number of samples.

(2) Spectral angle mapper (SAM)

SAM is used to measure the difference between the spectra of two images, with smaller values indicating higher spectral similarity. Its calculation formula is as follows:

$$SAM = \arccos \left( \frac{\sum_{i=1}^{n} P_i T_i}{\sqrt{\sum_{i=1}^{n} P_i^2} \sqrt{\sum_{i=1}^{n} T_i^2}} \right) \tag{9}$$

Where $P_i$ and $T_i$ represent the values of the i-th band for the predicted and true spectra, respectively.

(3) Mean structural similarity index (mSSI)

The Mean Structural Similarity Index measures the structural similarity between images, with larger values indicating greater structural similarity. Its calculation formula is as follows:

$$SSI(x, y) = \frac{(2\mu_x\mu_y + C_1)(2\sigma_{xy} + C_2)}{(\mu_x^2 + \mu_y^2 + C_1)(\sigma_x^2 + \sigma_y^2 + C_2)} \tag{10}$$

where x and y are the two images being compared, $\mu_x$ and $\mu_y$ are the average values of images x and y, $\sigma_x^2$ and $\sigma_y^2$ are the variances of images x and y, $\sigma_{xy}$ is the covariance between images x and y, and $C_1$ and $C_2$ are constants.
When calculating the Mean Structural Similarity Index, the SSIM values of all local windows are averaged:

$$mSSI = \frac{1}{M} \sum_{j=1}^{M} SSI(x_j, y_j) \tag{11}$$

where M is the total number of local windows.

(4) Correlation coefficient (CC)

The correlation coefficient is used to measure the strength and direction of the linear relationship between two variables, with larger values indicating a stronger correlation. The calculation formula is as follows:

$$CC = \frac{\sum_{i=1}^{n} (P_i - \overline{P})(T_i - \overline{T})}{\sqrt{\sum_{i=1}^{n} (P_i - \overline{P})^2 \sum_{i=1}^{n} (T_i - \overline{T})^2}} \tag{12}$$

where $P_i$ and $T_i$ represent the predicted values and true values, respectively, while $\overline{P}$ and $\overline{T}$ denote the means of the predicted and true values, respectively. n is the number of samples.
In addition to the quantitative metrics mentioned above, this paper also employed visual evaluation methods to assess the actual cloud removal effect of the images. By comparing the visual similarity between the generated cloud-free images and the actual cloud-free images, we can more intuitively evaluate the model's performance in detailed restoration and image quality.

### 3.3 Results

**3.3.1 Cloud removal results for region A.** Fig 7 showed the cloud removal results for the area of Region A. As shown in Fig 7(b), the coarse cloud-free image demonstrated a good overall cloud removal effect. Comparing Fig 7(b) with Fig 7(c), it could be observed that the refined cloud-free image presented better spectral details and texture structure in the areas previously covered by clouds.

**3.3.2 Cloud removal results for region B.** Fig 8 showed the cloud removal results for Region B, corresponding to Area 1 in Fig 6(d). Since no true cloud-free image was available for the validation experiment, the neighboring optical image (Fig 8(d)) was used as a reference for comparison. In the real cloud-covered scene, both the coarse cloud-free image (Fig 8(b)) and the refined cloud-free image (Fig 8(c)) demonstrated good cloud removal performance, with the resulting images being overall coherent and natural. For the local areas previously covered by clouds, the refined cloud-free image exhibited better spectral details and texture structure compared to the coarse cloud-free image.

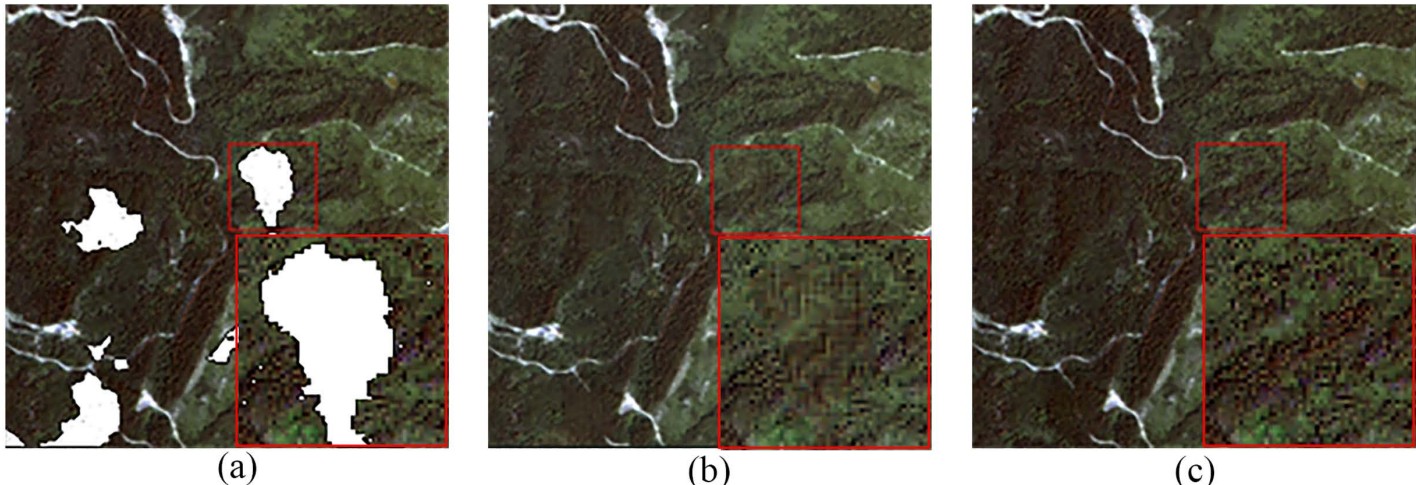

**Fig 7. Cloud Removal Results for Region A.** (a) Simulated Cloudy Image (b) Coarse Cloud Removal Image (c) Refined Cloud Removal Image.

## 3.4 Comparison of methods

To demonstrate the effectiveness of the cloud removal method proposed in this paper, a comparison was made between our method and the original CGAN [35], the deep learning-based method Simulation-Fusion GAN [20], and the Spatial Attention GAN [36]. The original CGAN took SAR images as input and used the CGAN model to generate optical RS images, thereby restoring the cloud-covered optical RS images. Simulation-Fusion GAN generated a relatively rough simulated optical image using SAR images, which was then fused with the cloud-covered optical image to achieve cloud removal. This method fully utilized the relationship between SAR and cloud-covered optical RS images, while also incorporating cloud-free spectral information from the optical RS images. The Spatial Attention GAN introduced a spatial attention mechanism into the cloud removal task, utilizing the local-to-global spatial attention mechanism to identify the cloud-covered areas and generate cloud-free images.

**3.4.1 Comparison of results in region A.** Table 1 presents a comparison of the four methods based on four quantitative metrics. As shown in Table 1, the method proposed in this paper outperforms the other three methods across all metrics. As illustrated in Fig 9, all four methods achieve good cloud removal, but there are differences in detail restoration. The original CGAN (Fig 9(b)) only generates partial roads, with noticeable blurred areas remaining in the cloud-free image. The Simulation-Fusion GAN (Fig 9(c)) not only performs poorly in spectral information recovery but also exhibits significant distortion in details. The Spatial Attention GAN (Fig 9(d)) performs well in detail restoration but suffers from excessive color enhancement, with overly bright yellow regions and inaccurate spectral information recovery. In contrast, the method proposed in this paper (Fig 9(e)) shows superior performance in all three aspects, effectively removing clouds and accurately restoring spectral and texture information, resulting in a better overall visual effect. Overall, the method proposed in this paper outperforms the original CGAN, Simulation-Fusion GAN, and Spatial Attention GAN in terms of cloud removal effectiveness under simulated cloudy conditions.

**3.4.2 Comparison results in region B.** According to the cloud detection model Refined U-Net [27], the cloud coverage in the optical RS image of Region B (Fig 6(d)) accounts for 24.81% of the total area. Three regions were selected for cloud removal effect comparison, as shown in Figs 10, 11, and 12, which correspond to regions 2, 3, and 4 in Fig 6(d), respectively. These three regions represent different scenes. Region 2 has a relatively flat terrain, with a larger area of cloud shadow coverage. The original CGAN result (Fig 10(b)) shows noticeable cloud shadow remnants and does not effectively remove the cloud shadow. Region 3 is a typical mountainous terrain, focusing on the connection

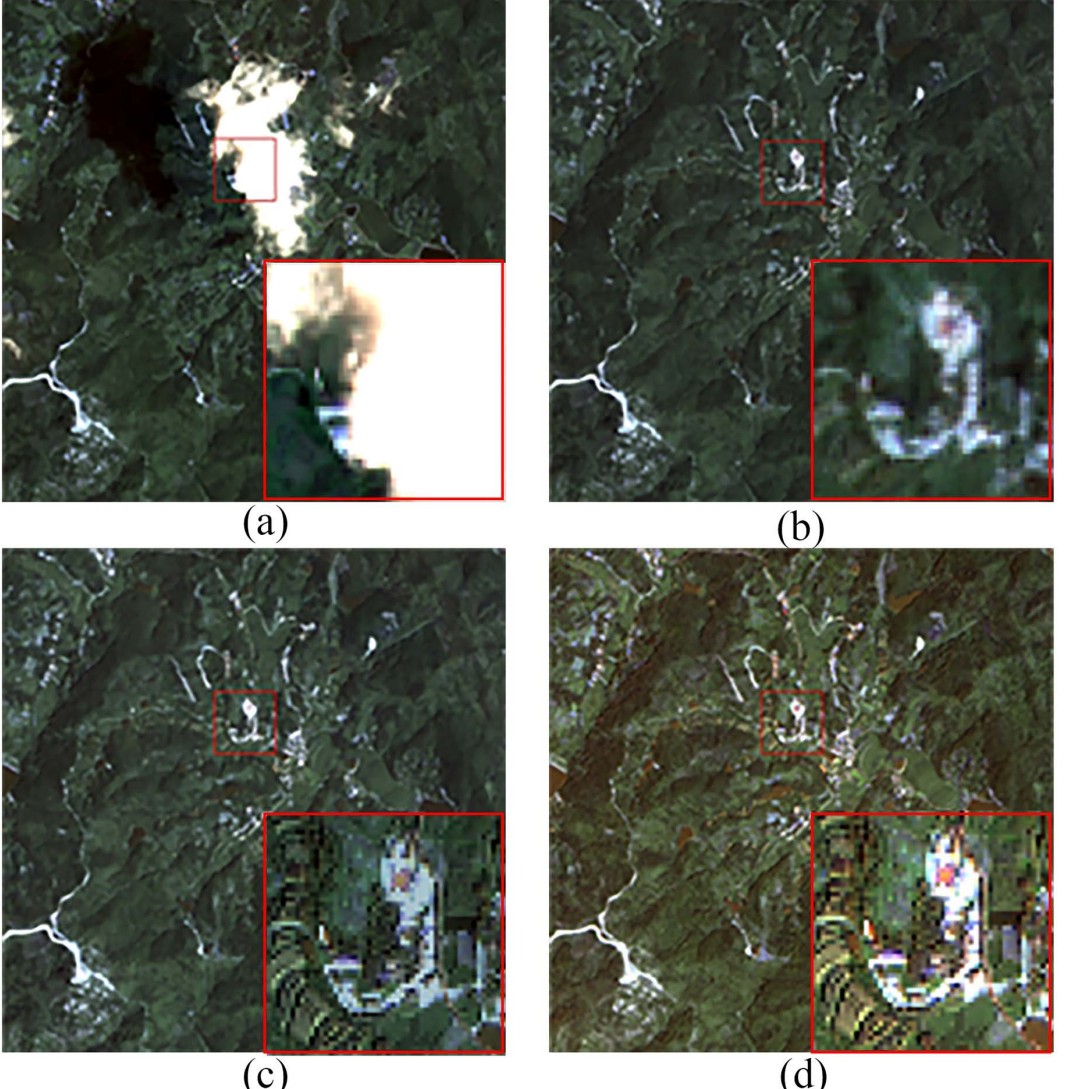

**Fig 8. Cloud Removal Results for Region B.** (a) Original Cloudy Image (b) Coarse Cloud Removal Image (c) Refined Cloud Removal Image (d) Neighboring Optical Image.

**Table 1. Quantitative Evaluation of Comparison Methods on Region A.**

| Method | RMSE | SAM | mSSI | CC |
|---|---|---|---|---|
| Proposed Method | **0.0391** | **0.0729** | **0.9221** | **0.9537** |
| Original CGAN | 0.1083 | 0.1714 | 0.8515 | 0.9054 |
| Simulation-Fusion GAN | 0.0761 | 0.1337 | 0.8872 | 0.9190 |
| Spatial Attention GAN | 0.0510 | 0.1167 | 0.9004 | 0.9297 |

between the cloudy and cloud-free areas. The results of the original CGAN (Fig 11(b)) and Simulation-Fusion GAN (Fig 11(c)) show obvious boundary traces. Although the Spatial Attention GAN result (Fig 11(d)) lacks boundary traces, there is a severe loss of spatial texture information compared to the neighboring optical image (Fig 11(f)). Region 4 is a

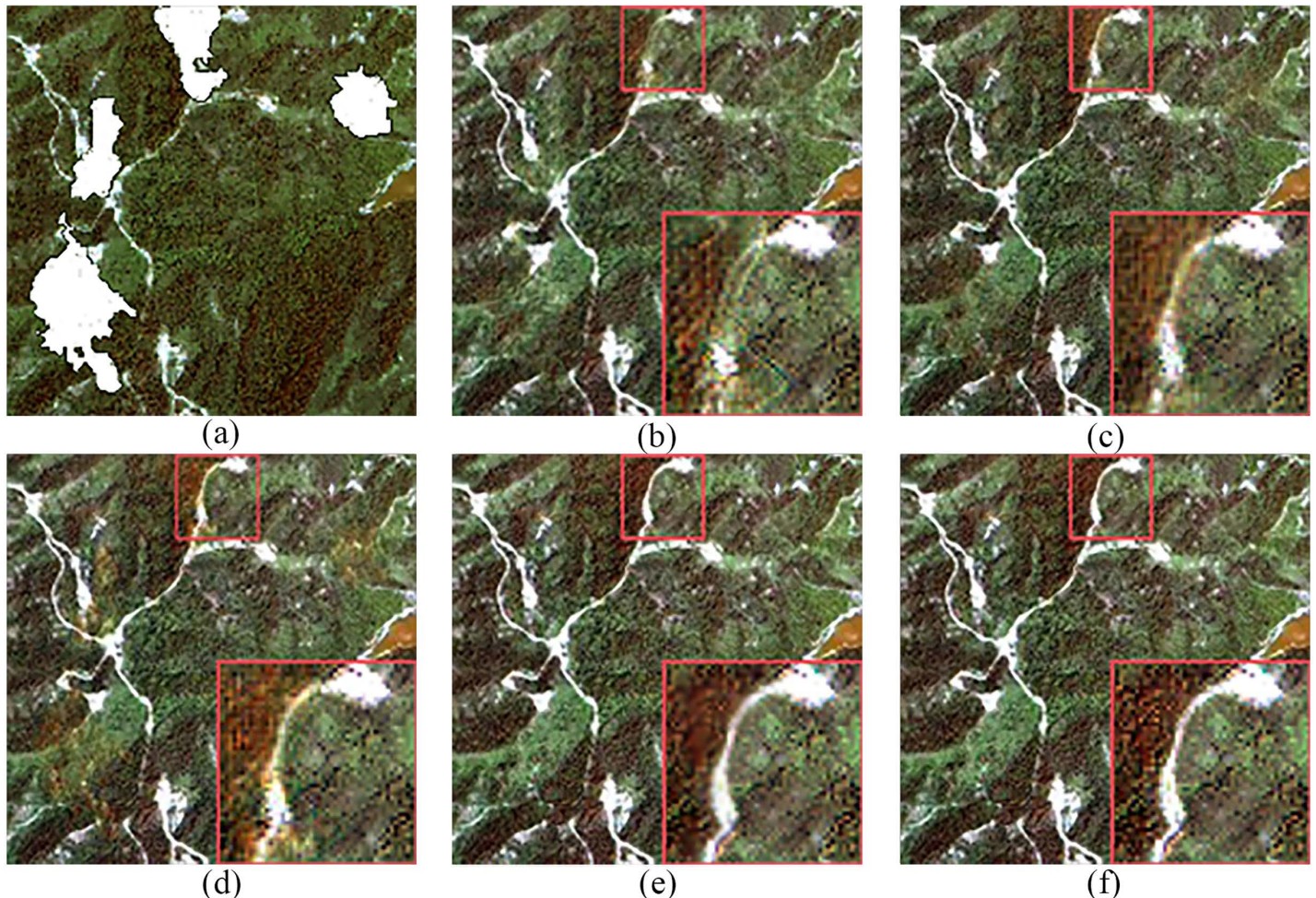

**Fig 9. Results of Simulated Experiment in Local Areas.** (a) Simulated Cloudy Image (b) Original CGAN (c) Simulation-Fusion GAN (d) Spatial Attention GAN (e) Proposed Method (f) Actual Cloud-Free Image.

relatively complex urban area, which tests the model's restoration ability in complex scenes. The results of the original CGAN, Simulation-Fusion GAN, and Spatial Attention GAN (Figs 12(b), 12(c), 12(d)) show detail loss and blurred textures compared to the neighboring optical image (Fig 12(f)), with poor cloud removal effects in complex landscape areas. In contrast, the method proposed in this paper effectively removes clouds and cloud shadows, with a natural transition in the connection parts. The recovery of spatial details in complex landscape areas is outstanding, showing the best cloud removal performance.

## 4 Discussion

### 4.1 Feature ablation experiments

As we demonstrated in Section 2, the proposed coarse-to-fine cloud removal method for optical RS images utilizes four types of feature information: SAR image local features, SAR image global features, optical RS image edge features, and local spectral features of neighboring optical RS images. To evaluate the contribution of each feature, we designed four ablation experiments by removing one feature while retaining the others. These experiments are: (1) without SAR image

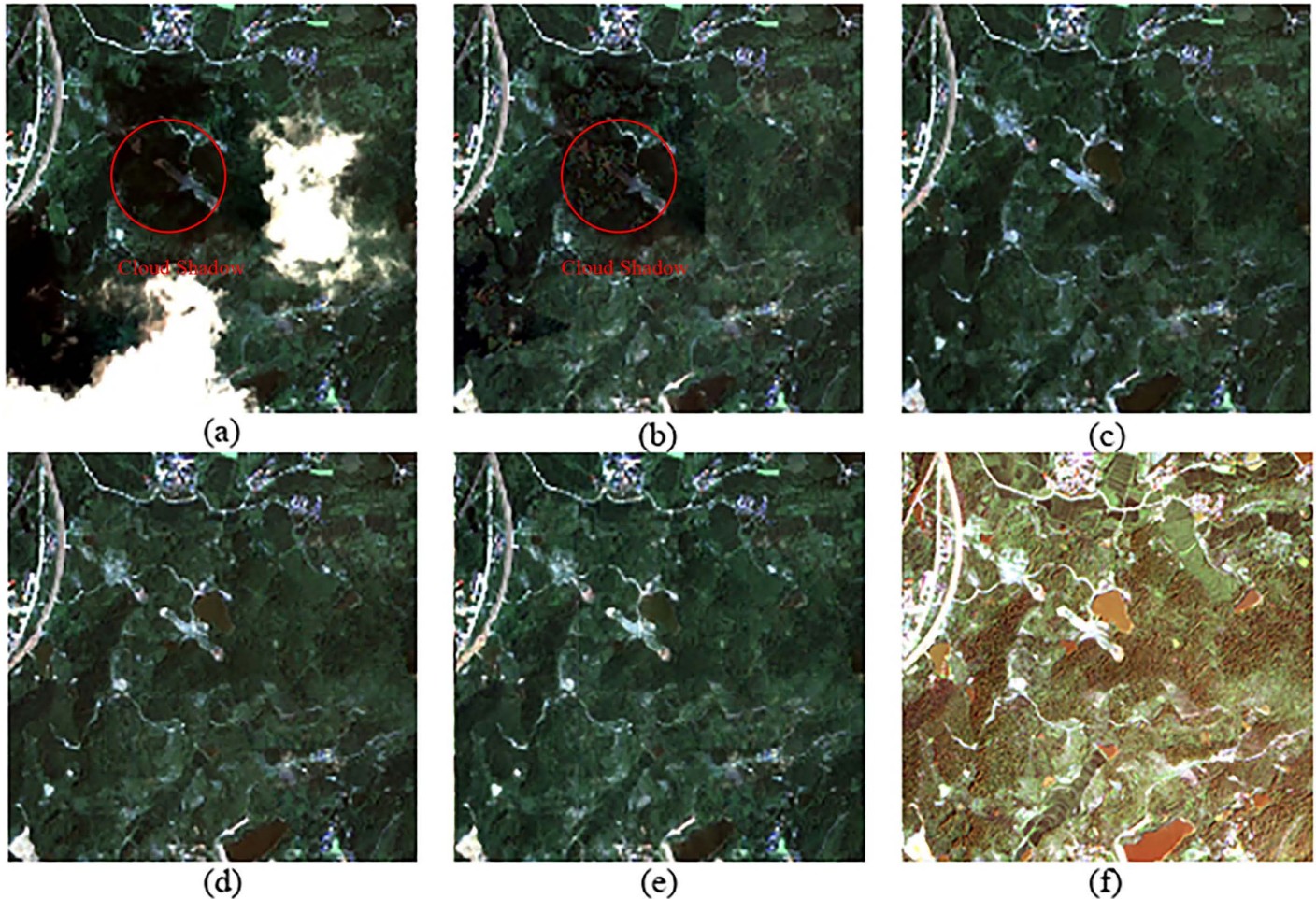

**Fig 10. Cloud Removal Comparison for Region 2 in Area B.** (a) Original Cloudy Image (b) Original CGAN (c) Simulation-Fusion GAN (d) Spatial Attention GAN (e) Proposed Method (f) Neighboring Optical RS Image.

local spectral features, (2) without SAR image global spectral features, (3) without optical RS image edge features, and (4) without local spectral features of neighboring optical RS images. The results of these four ablation experiments are compared with the results of the proposed method. Quantitative evaluation metrics for the feature ablation experiments were conducted using Region A data, as shown in Table 2. Additionally, visual cloud removal effect comparisons were made using Region 5 of Region B data, as shown in Fig 13.

From Table 2, we can observe that the results of experiments without SAR image local features and without local spectral features of neighboring optical RS images show significant degradation compared to the proposed method in various aspects. When SAR image local features are absent, all evaluation metrics except the Spectral Angle Mapper (SAM) are at their lowest, and the cloud-free image exhibits poor texture detail recovery. This indicates the important role that SAR image local features play in restoring image details. When the local spectral features of neighboring optical RS images are absent, the SAM decreases significantly, reflecting poor spectral accuracy. This shows that neighboring optical RS images provide critical spectral information for the cloud-covered areas. The cases without optical RS image edge features and SAR image local features both exhibit declines in various metrics compared to the proposed method, indicating that adding these two features enhances the model's cloud removal performance. Therefore, from the quantitative analysis

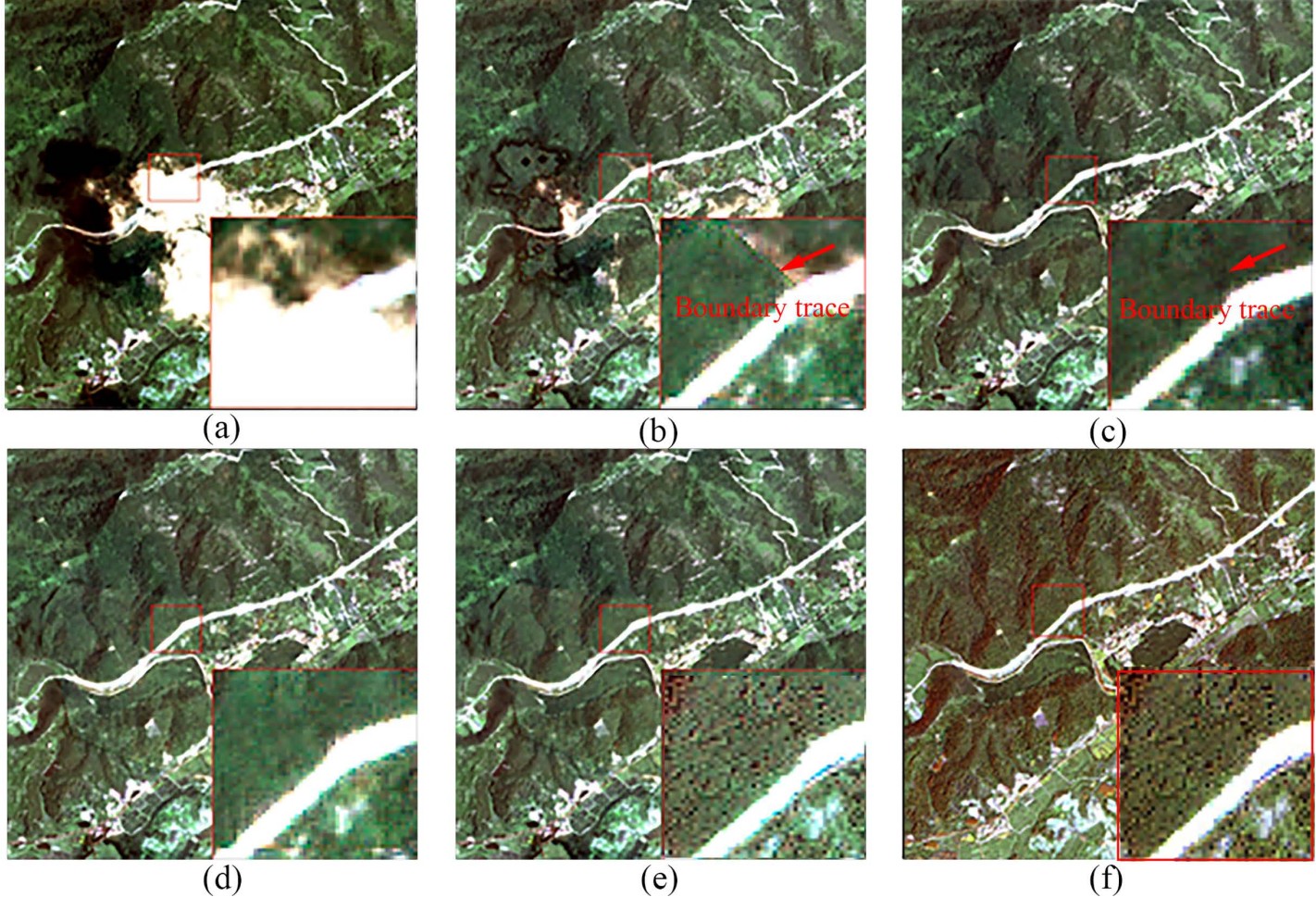

**Fig 11. Cloud Removal Comparison for Region 3 in Area B.** (a) Original Cloudy Image (b) Original CGAN (c) Simulation-Fusion GAN (d) Spatial Attention GAN (e) Proposed Method (f) Neighboring Optical RS Image.

perspective, the proposed method effectively integrates the advantages of all four features, achieving the best performance in terms of spectral accuracy and texture details, resulting in high precision.

From Fig 13, it is clear that the cases without SAR image local features (Fig 13 (b)) and without local spectral features of neighboring optical RS images (Fig 13 (c)) show evident deficiencies in cloud removal effectiveness. The absence of SAR image local features leads to acceptable spectral information restoration but poor detail recovery. For example, in the region where the lake and land meet (highlighted by the red arrow), the boundary that should be clear becomes blurry, lacking distinct separation. This phenomenon indicates that the absence of SAR image local features causes problems in restoring image details and fails to reflect the depth and detail of the real scene. On the other hand, the absence of local spectral features of neighboring optical RS images results in better structural and detail restoration but notable spectral issues. The image overall appears too dark, with colors lacking depth and vibrancy. This highlights the importance of local spectral features from neighboring optical RS images in restoring spectral information. In contrast, the proposed method (Fig 13 (d)), which integrates SAR image features and neighboring optical RS image features, generates cloud-free images that achieve a good balance between spectral restoration and detail recovery, with a complete image structure, demonstrating the effectiveness of the method.

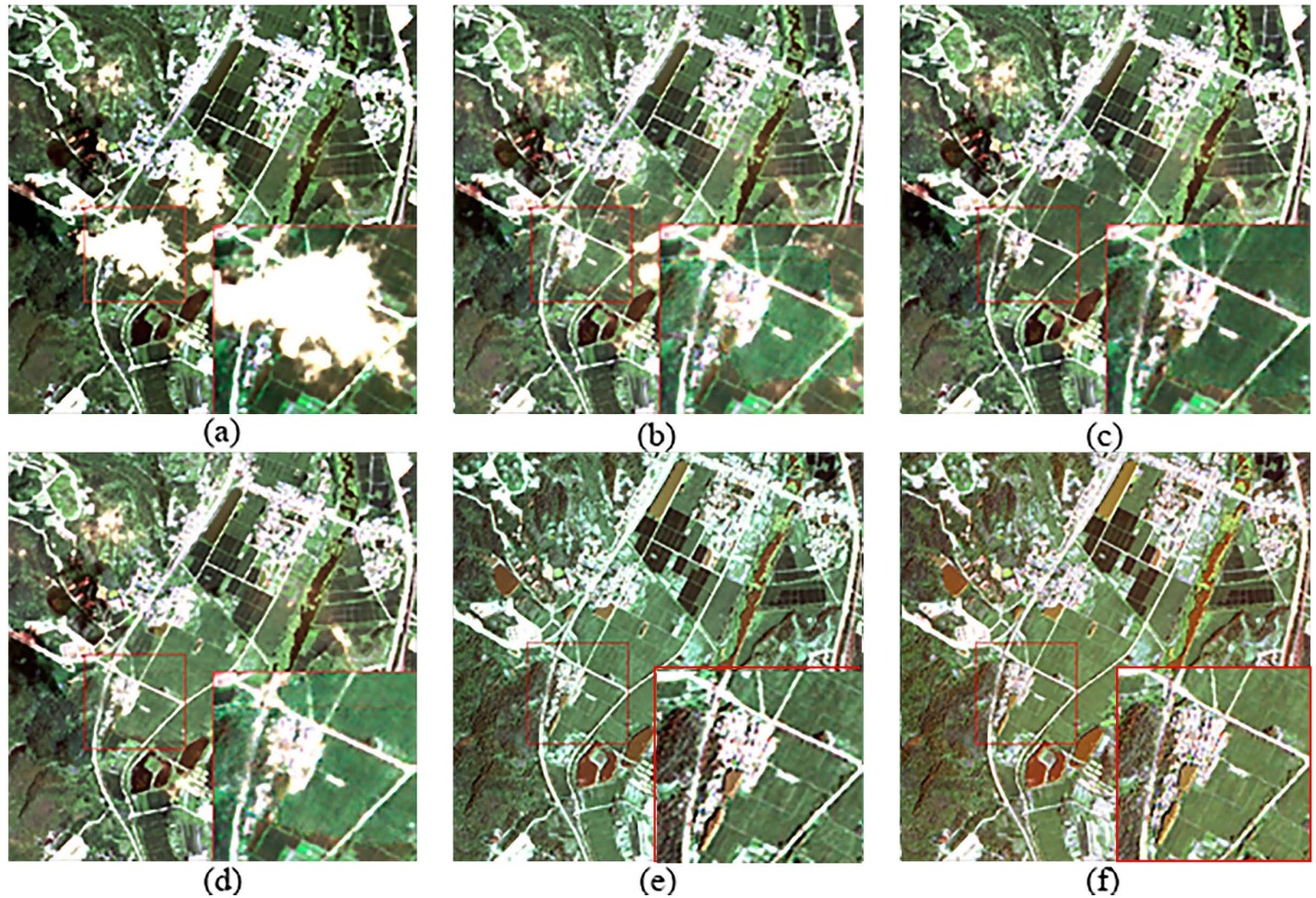

**Fig 12. Cloud Removal Comparison for Region 4 in Area B.** (a) Original Cloudy Image (b) Original CGAN (c) Simulation-Fusion GAN (d) Spatial Attention GAN (e) Proposed Method (f) Neighboring Optical RS Image.

**Table 2. Quantitative Evaluation of Feature Ablation Experiment.**

|  | RMSE | SAM | mSSI | CC |
|---|---|---|---|---|
| Proposed Method | **0.0391** | **0.0729** | **0.9221** | **0.9537** |
| Without SAR Image Local Spectral Features | 0.1961 | 0.1360 | 0.8214 | 0.8119 |
| Without SAR Image Global Spectral Features | 0.0533 | 0.0914 | 0.8921 | 0.9195 |
| Without Optical RS Image Edge Features | 0.0673 | 0.1112 | 0.8615 | 0.9213 |
| Without Local Spectral Features of Neighboring Optical RS Images | 0.1834 | 0.2277 | 0.8560 | 0.8508 |

## 4.2 Ablation experiment of CGAN model

To optimize the performance of the CGAN model for cloud removal tasks, we conducted ablation experiments on the number of layers in both the generator and discriminator networks to examine the impact of network depth on model performance and determine the optimal network structure. In the generator ablation experiment, we set the discriminator network to 4 layers and gradually increased the number of layers in the generator network from 8 to 16 layers. In

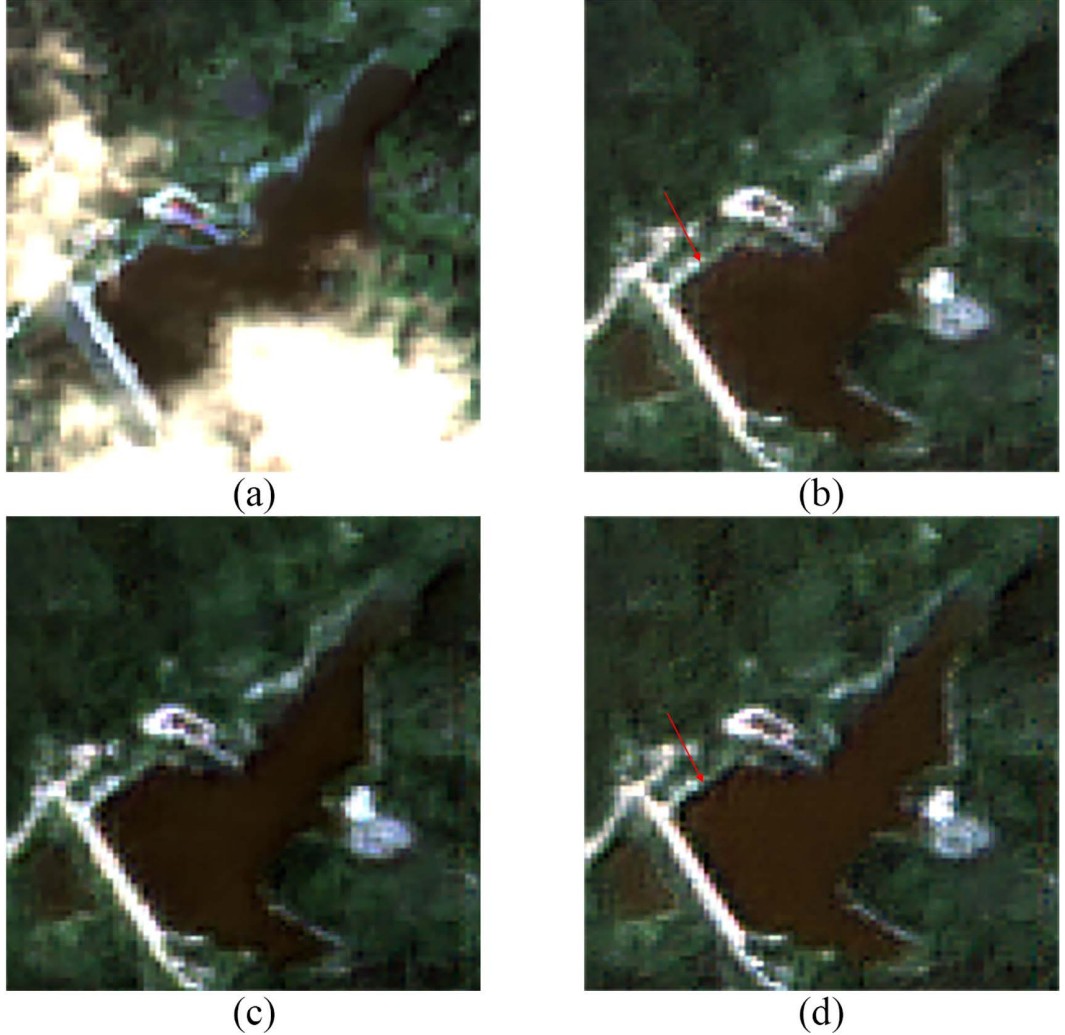

**Fig 13. Visual Effect Evaluation of Feature Ablation Experiment.** (a) Original Cloudy Image (b) Without SAR Image Features (c) Without Neighboring Optical RS Features (d) Proposed Method.

the discriminator ablation experiment, we fixed the generator network at 6 layers and gradually increased the number of layers in the discriminator network from 2 to 8 layers. We used the Root Mean Square Error (RMSE) as the evaluation metric to quantify the difference between the cloud-free image generated by the model and the real cloud-free image.

The experimental results are shown in Fig 14. As the depth of the generator network increases, the RMSE metric gradually decreases, demonstrating significant performance improvement. When the generator network reaches 14 layers, the RMSE achieves its optimal value. Further increasing the number of layers leads to longer training times without improving the RMSE, so 14 layers are determined to be the optimal structure for the generator network. When the discriminator network reaches 4 layers, the RMSE achieves its optimal value of 0.0391, aligning with the best structure of the generator network, and the overall model performance reaches its peak. Beyond 4 layers, increasing the complexity of the discriminator does not yield further performance improvements, so 4 layers are selected as the optimal structure for the discriminator.

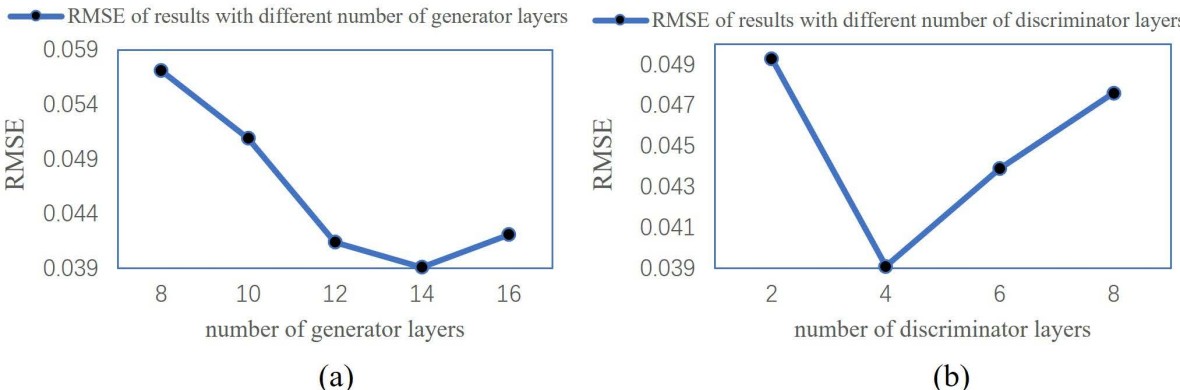

(a) (b)

**Fig 14. Ablation Study of Network Depth for CGAN Model.** (a) generator (b) discriminator.

After determining the optimal model structure, we conducted an ablation experiment on the composite loss function, which includes five cases: without L1 loss, without local loss, without GAN loss, without perceptual loss, and without edge loss. Using data from Region A, we compared the quantitative evaluation metrics of the ablation experiments on the composite loss function. The results of the five ablation experiments were compared with those of the proposed method, as shown in Table 3.

The results show that the case without L1 loss is very close to the results of the proposed method, but it cannot control the global distribution of the output. The results without local loss, GAN loss, and edge loss show a slight decrease in various metrics compared to the proposed method. From Table 3, it can be observed that the result without perceptual loss performs better in quantitative evaluation, but the perceptual loss function enriches the texture structure from a visual perspective, as reflected in the mSSI index. The experiments demonstrate that the proposed method, which combines multiple loss functions, achieves the best performance, providing a balanced result in both spectral restoration and texture detail.

After discussing the composite loss function, we determined the optimal values for the five parameters in the composite loss function: $\lambda_{CGAN}$, $\lambda_{L1}$, $\lambda_{perceptual}$, $\lambda_{edge}$, and $\lambda_{local}$, by adjusting the parameters to bring the model to its optimal state. Since there are five parameters and different loss terms may have coupling relationships, we adopted a strategy of gradually increasing the weight of each loss term in powers of 10. First, we fixed the weight of L1 loss ($\lambda_{L1}$) at 10, followed by determining the weight of local loss ($\lambda_{local}$), then the weights for perceptual loss ($\lambda_{perceptual}$) and edge loss ($\lambda_{edge}$), and finally, determining the weight of GAN loss ($\lambda_{CGAN}$). To quantify the evaluation of each loss term's weight, we used the RMSE metric to compare models with different weight values. The adjustment of the loss function parameters not only optimized the model's performance but also allowed for precise control over the influence of each loss term on the final result. Fig 15 illustrates the impact of different loss term parameters on the model results.

**Table 3. Quantitative Evaluation of the Ablation Experiment for Five Loss Functions.**

|  | RMSE | SAM | mSSI | CC |
|---|---|---|---|---|
| Proposed Method | 0.0391 | 0.0729 | **0.9221** | 0.9537 |
| Without L1 Loss | 0.0474 | 0.0789 | 0.9106 | 0.9438 |
| Without Local Loss | 0.0411 | 0.0814 | 0.9018 | 0.9363 |
| Without GAN Loss | **0.0351** | 0.0712 | 0.9137 | 0.9420 |
| Without Perceptual Loss | 0.0396 | 0.0831 | 0.9131 | **0.9549** |
| Without Edge Loss | 0.0368 | **0.0698** | 0.9027 | 0.9218 |

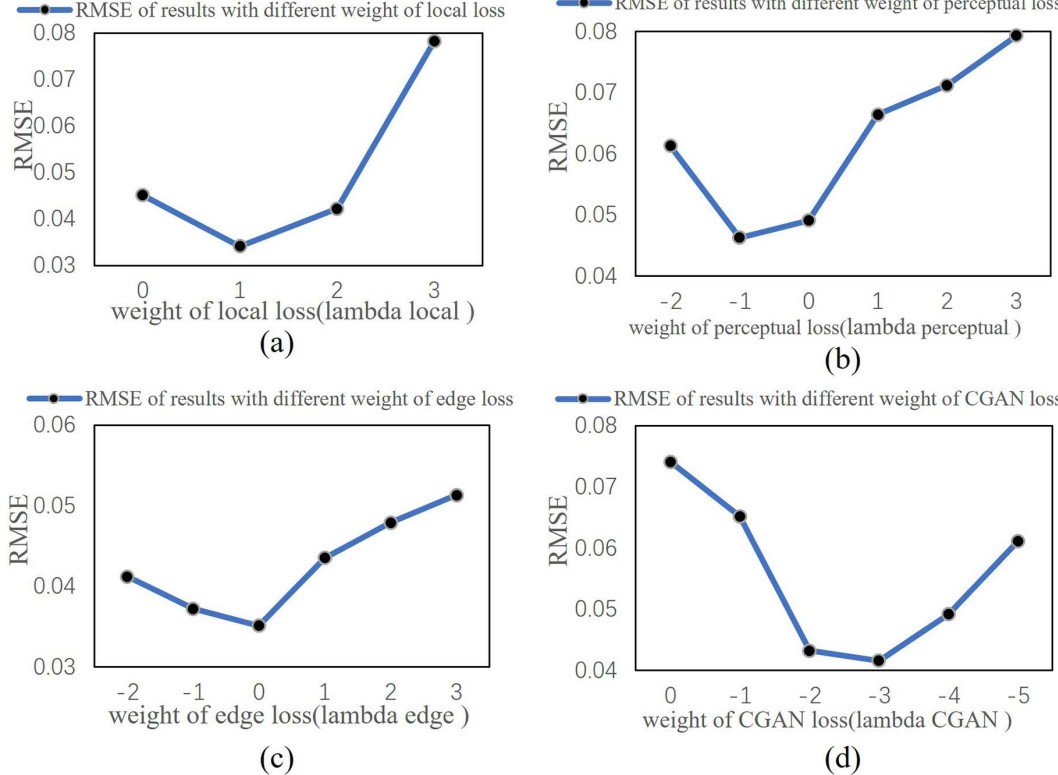

**Fig 15. Ablation Study of Compound Loss Function Parameters in CGAN Model.** (a) $\lambda_{local}$ (b) $\lambda_{perceptual}$ (c) $\lambda_{edge}$ (d) $\lambda_{CGAN}$.

### 4.3 Applicability

The overall process of this study involves using SAR images for coarse cloud removal on cloud-covered optical images, followed by fine cloud removal using neighboring optical RS images. The applicability of the method is demonstrated in the following aspects:

(1) SAR images are easily accessible.

SAR images are not affected by weather conditions, making data acquisition relatively easy. Experimental results have shown that using SAR images for coarse cloud removal on optical RS images yields satisfactory results.

(2) Neighboring optical RS images are readily available.

In this study, neighboring optical RS images refer to cloud-free optical images from the same region as the cloud-covered optical image. These images can be from the same sensor at different times or from different sensors covering the same region. Fig 16 shows a comparison of cloud removal effects using different neighboring optical RS images. A comparison between Fig 16(b) and 16(c) reveals that neighboring optical images from different sensors can still achieve good cloud removal. This indicates that if neighboring optical images from the same sensor at different times are not available, using images from other sensors for the same region can still demonstrate good applicability.

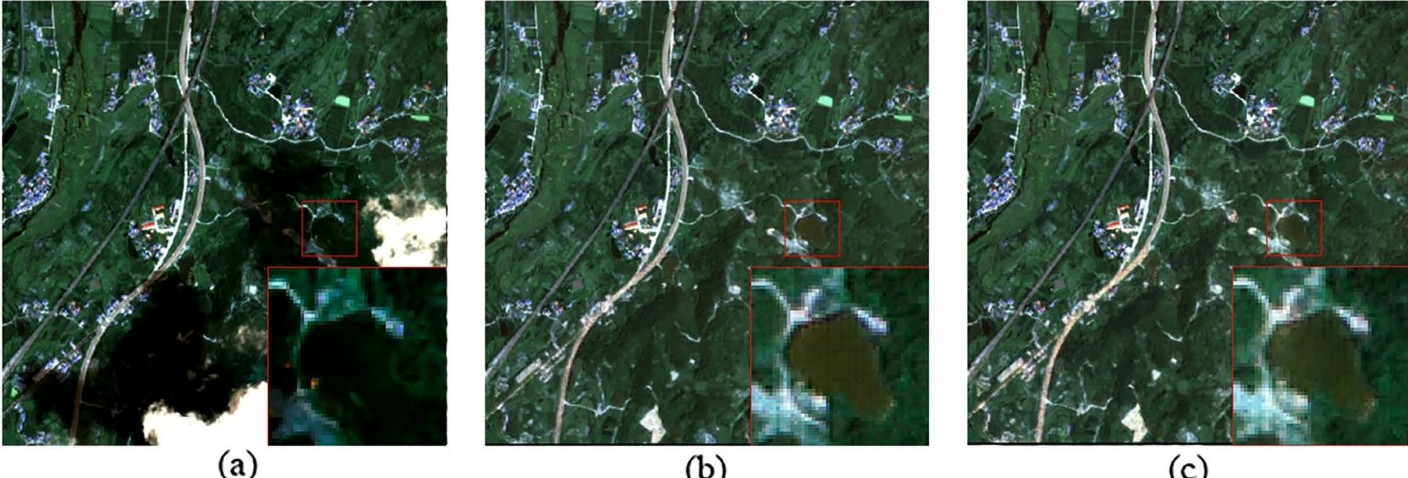

**Fig 16. Comparison of cloud removal effects using different neighboring optical RS images.** (a) Cloud-covered optical RS image; (b) Cloud removal effect using GF-2 neighboring optical RS image; (c) Cloud removal effect using Sentinel-2 neighboring optical RS image.

## 4.4 Limitations and prospects

During the testing process of the proposed method, some limitations were still identified, underscoring the need for further research.

1) Model generalization ability. Currently, the model was applied to optical RS images in the Dabie Mountains, China, and achieved satisfactory cloud removal results. However, it has not been extensively tested in other regions. Differences in land cover types, spectral characteristics, and climatic conditions across different regions may lead to performance degradation when the model is directly applied to untrained areas. Therefore, future research will explore strategies based on transfer learning or adaptive learning to enhance the model's generalization ability, ensuring stable cloud removal performance under broader geographical and climatic conditions.

2) Further improvement of cloud removal performance. Extensive experiments have demonstrated the strong applicability of the proposed method. Further enhancement of cloud removal performance could be achieved through the following two aspects: i) Accurate registration of optical RS and SAR images. The registration accuracy directly affects the effectiveness of feature extraction, thereby influencing the cloud removal results. However, precise registration between optical and SAR images remains a challenging research topic in RS. More optimized registration methods could be explored to facilitate cloud removal in optical RS images. ii) Selection of the optimal neighboring optical RS image. Theoretically, the proposed method does not impose specific requirements on the neighboring optical RS image. In practice, differences in sensor types, spatial resolution, spectral characteristics, and acquisition time may affect the accuracy of cloud removal. Future research could investigate the impact of these factors on cloud removal performance and attempt to establish criteria for selecting the optimal neighboring optical RS image.

3) Evaluation of the application potential of cloud-free optical RS images. The current study mainly focused on cloud removal effectiveness, without further assessing the performance of the cloud-free images in downstream applications. For instance, cloud-free images could be used for land cover classification, but their reliability and applicability still need further verification. Future research could integrate cloud-free images into land cover classification tasks to evaluate the impact of cloud removal on classification accuracy, thereby comprehensively validating and assessing the practical value and application potential of the proposed method.

## 5 Conclusion

This paper proposes a coarse-to-fine cloud removal method for optical RS images. Compared to other methods, the proposed approach demonstrates better cloud removal performance and excellent applicability. The specific conclusions are as follows:

[1] Based on the deep CGAN network, combining SAR images with an edge-constrained strategy enables effective coarse cloud removal on optical RS images. The use of neighboring optical RS images further refines the cloud removal effect on optical images.

[2] Compared with three other common cloud removal methods, the proposed method not only achieves better cloud removal accuracy (RMSE = 0.0238, SAM = 0.0629, mSSI = 0.9421, CC = 0.9737) but also provides the best visual quality of the cloud-free images.

[3] The SAR images used in this method are easily accessible, and neighboring optical RS images are readily available, making the method highly applicable.

## Author contributions

**Methodology:** Yuyao Wang, Jiehai Cheng.

**Writing – original draft:** Yuyao Wang, Jiehai Cheng.

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
