## [Decision Letter · Decision Letter 0]

20 Feb 2025

Dear Dr. Cheng,

Thank you for submitting your manuscript to PLOS ONE. After careful consideration, we feel that it has merit but does not fully meet PLOS ONE’s publication criteria as it currently stands. Therefore, we invite you to submit a revised version of the manuscript that addresses the points raised during the review process.

**The manuscript must be corrected in all points indicated by the reviewers, such as:**

**1. The abstract should include a reflection of the specific indicators and numerical values of the experimental results;**

2. It would be very useful if this algorithm is applied to a large area, for example an area of 30 km by 30 km;

3. Before introducing the contribution of this article, provide a more clear transition;

4. Restructure Key Contributions: It may be worth condensing these points into clear contributions and then elaborate on them further in the document;

5. The clarity of the experimental results figures in the article is poor. Improve the figures quality appropriately;

6. It is necessary to discuss the unpredictable results or patterns encountered in this study, as well as the limitations of this research and draw some future directions.

We look forward to receiving your revised manuscript.

Kind regards,

Claudionor Ribeiro da Silva

Academic Editor

PLOS ONE

**Journal Requirements:**

Please ensure that your manuscript meets PLOS ONE's style requirements, including those for file naming. The PLOS ONE style templates can be found at https://journals.plos.org/plosone/s/file?id=wjVg/PLOSOne_formatting_sample_main_body.pdf and https://journals.plos.org/plosone/s/file?id=ba62/PLOSOne_formatting_sample_title_authors_affiliations.pdf 2. Please note that PLOS ONE has specific guidelines on code sharing for submissions in which author-generated code underpins the findings in the manuscript. In these cases, we expect all author-generated code to be made available without restrictions upon publication of the work. Please review our guidelines at https://journals.plos.org/plosone/s/materials-and-software-sharing#loc-sharing-code and ensure that your code is shared in a way that follows best practice and facilitates reproducibility and reuse. 3. We note that the grant information you provided in the ‘Funding Information’ and ‘Financial Disclosure’ sections do not match.  When you resubmit, please ensure that you provide the correct grant numbers for the awards you received for your study in the ‘Funding Information’ section. 4. We note that your Data Availability Statement is currently as follows: All relevant data are within the manuscript and its Supporting Information files. Please confirm at this time whether or not your submission contains all raw data required to replicate the results of your study. Authors must share the “minimal data set” for their submission. PLOS defines the minimal data set to consist of the data required to replicate all study findings reported in the article, as well as related metadata and methods (https://journals.plos.org/plosone/s/data-availability#loc-minimal-data-set-definition). For example, authors should submit the following data: - The values behind the means, standard deviations and other measures reported;- The values used to build graphs;- The points extracted from images for analysis. Authors do not need to submit their entire data set if only a portion of the data was used in the reported study. If your submission does not contain these data, please either upload them as Supporting Information files or deposit them to a stable, public repository and provide us with the relevant URLs, DOIs, or accession numbers. For a list of recommended repositories, please see https://journals.plos.org/plosone/s/recommended-repositories. If there are ethical or legal restrictions on sharing a de-identified data set, please explain them in detail (e.g., data contain potentially sensitive information, data are owned by a third-party organization, etc.) and who has imposed them (e.g., an ethics committee). Please also provide contact information for a data access committee, ethics committee, or other institutional body to which data requests may be sent. If data are owned by a third party, please indicate how others may request data access. 5. We note that Figure 6 in your submission contain map images which may be copyrighted. All PLOS content is published under the Creative Commons Attribution License (CC BY 4.0), which means that the manuscript, images, and Supporting Information files will be freely available online, and any third party is permitted to access, download, copy, distribute, and use these materials in any way, even commercially, with proper attribution. For these reasons, we cannot publish previously copyrighted maps or satellite images created using proprietary data, such as Google software (Google Maps, Street View, and Earth). For more information, see our copyright guidelines: http://journals.plos.org/plosone/s/licenses-and-copyright. We require you to either present written permission from the copyright holder to publish these figures specifically under the CC BY 4.0 license, or remove the figures from your submission: a. You may seek permission from the original copyright holder of Figure 6 to publish the content specifically under the CC BY 4.0 license.   We recommend that you contact the original copyright holder with the Content Permission Form (http://journals.plos.org/plosone/s/file?id=7c09/content-permission-form.pdf) and the following text:“I request permission for the open-access journal PLOS ONE to publish XXX under the Creative Commons Attribution License (CCAL) CC BY 4.0 (http://creativecommons.org/licenses/by/4.0/). Please be aware that this license allows unrestricted use and distribution, even commercially, by third parties. Please reply and provide explicit written permission to publish XXX under a CC BY license and complete the attached form.” Please upload the completed Content Permission Form or other proof of granted permissions as an "Other" file with your submission. In the figure caption of the copyrighted figure, please include the following text: “Reprinted from [ref] under a CC BY license, with permission from [name of publisher], original copyright [original copyright year].” b. If you are unable to obtain permission from the original copyright holder to publish these figures under the CC BY 4.0 license or if the copyright holder’s requirements are incompatible with the CC BY 4.0 license, please either i) remove the figure or ii) supply a replacement figure that complies with the CC BY 4.0 license. Please check copyright information on all replacement figures and update the figure caption with source information. If applicable, please specify in the figure caption text when a figure is similar but not identical to the original image and is therefore for illustrative purposes only.The following resources for replacing copyrighted map figures may be helpful: USGS National Map Viewer (public domain): http://viewer.nationalmap.gov/viewer/The Gateway to Astronaut Photography of Earth (public domain): http://eol.jsc.nasa.gov/sseop/clickmap/Maps at the CIA (public domain): https://www.cia.gov/library/publications/the-world-factbook/index.html and https://www.cia.gov/library/publications/cia-maps-publications/index.htmlNASA Earth Observatory (public domain): http://earthobservatory.nasa.gov/Landsat:
http://landsat.visibleearth.nasa.gov/USGS EROS (Earth Resources Observatory and Science (EROS) Center) (public domain): http://eros.usgs.gov/#Natural Earth (public domain): http://www.naturalearthdata.com/ 6. We note that Figures 7, 8, 9, 10, 11, 12, 13 and 16 in your submission contain copyrighted images. All PLOS content is published under the Creative Commons Attribution License (CC BY 4.0), which means that the manuscript, images, and Supporting Information files will be freely available online, and any third party is permitted to access, download, copy, distribute, and use these materials in any way, even commercially, with proper attribution. For more information, see our copyright guidelines: http://journals.plos.org/plosone/s/licenses-and-copyright. We require you to either present written permission from the copyright holder to publish these figures specifically under the CC BY 4.0 license, or remove the figures from your submission: a. You may seek permission from the original copyright holder of Figures 7, 8, 9, 10, 11, 12, 13 and 16 to publish the content specifically under the CC BY 4.0 license.  We recommend that you contact the original copyright holder with the Content Permission Form (http://journals.plos.org/plosone/s/file?id=7c09/content-permission-form.pdf) and the following text:“I request permission for the open-access journal PLOS ONE to publish XXX under the Creative Commons Attribution License (CCAL) CC BY 4.0 (http://creativecommons.org/licenses/by/4.0/). Please be aware that this license allows unrestricted use and distribution, even commercially, by third parties. Please reply and provide explicit written permission to publish XXX under a CC BY license and complete the attached form.” Please upload the completed Content Permission Form or other proof of granted permissions as an "Other" file with your submission.  In the figure caption of the copyrighted figure, please include the following text: “Reprinted from [ref] under a CC BY license, with permission from [name of publisher], original copyright [original copyright year].” b. If you are unable to obtain permission from the original copyright holder to publish these figures under the CC BY 4.0 license or if the copyright holder’s requirements are incompatible with the CC BY 4.0 license, please either i) remove the figure or ii) supply a replacement figure that complies with the CC BY 4.0 license. Please check copyright information on all replacement figures and update the figure caption with source information. If applicable, please specify in the figure caption text when a figure is similar but not identical to the original image and is therefore for illustrative purposes only.

Reviewers' comments:

Reviewer's Responses to Questions

**Comments to the Author**

1. Is the manuscript technically sound, and do the data support the conclusions?

Reviewer #1: Yes

Reviewer #2: Yes

2. Has the statistical analysis been performed appropriately and rigorously?

Reviewer #1: Yes

Reviewer #2: Yes

3. Have the authors made all data underlying the findings in their manuscript fully available?

Reviewer #1: Yes

Reviewer #2: No

4. Is the manuscript presented in an intelligible fashion and written in standard English?

Reviewer #1: Yes

Reviewer #2: Yes

**Reviewer #1:**  The manuscript entitled "A novel cloud removal method by fusing features from SAR and neighboring optical remote sensing images" aimed to propose an innovative cloud removal technique that integrates characteristics from Synthetic Aperture Radar (SAR) and adjacent optical remote sensing images in a small part of Dabie Mountains, China.

The manuscript is interesting and well structured. I believe it is suitable to be published in "PLOS" after addressing the following points.

1) Page 22, Datasets. It would be very useful if this algorithm is applied to a large area, for example an area of 30 km by 30 km. Can you do this experiment?

2) Page 32, Discussion. Discuss the limitations of your method and draw some future directions.

**Reviewer #2:**  This paper proposes a novel cloud cleaning method based on deep CGAN network that integrates SAR and neighboring optical remote sensing image features. The content of the article is complete, the experiments are sufficient, and the conclusions are well proven. However, there are some issues that the author needs to address in order to further improve the quality of the article:

1. The abstract should include a reflection of the specific indicators and numerical values of the experimental results;

2. At the end of the abstract, a sentence should be provided stating the significance of this study for the field;

3. The sentence 'It can be confidently stated that' in line 97 of the introduction is not suitable for use in the paper. Please cite appropriate references as support;

4. Before introducing the contribution of this article, provide a more clear transition;

5. Restructure Key Contributions: It may be worth condensing these points into clear contributions and then elaborate on them further in the document.

6. It is necessary to improve the resolution of Figure 6. In addition, the clarity of the experimental results figures in the article is poor. Please improve the figures quality appropriately�you can convert the image into a vector image in SVG format;

7. It is necessary to discuss the unpredictable results or patterns encountered in this study, as well as the limitations of this research.

**Do you want your identity to be public for this peer review?** For information about this choice, including consent withdrawal, please see our Privacy Policy

Reviewer #1: No

Reviewer #2: No

---

## [Author Response · Author response to Decision Letter 1]

19 Mar 2025

Dear Editor and Reviewers,

We sincerely appreciate your positive and constructive comments on our paper. Based on the suggestions from the editor and reviewers, we have carefully revised the manuscript and marked the changes using the track changes mode.

Below, we provide a detailed list of the revisions made compared to the original submission. We are truly grateful for all the valuable feedback from the reviewers and the editor.

Paper: A novel cloud removal method by fusing features from SAR and neighboring optical remote sensing images

Response to the Editor:

Response to the Editor's Comments

1. When submitting your revision, we need you to address these additional requirements. Please ensure that your manuscript meets PLOS ONE's style requirements, including those for file naming.

Response: We adjusted the manuscript to comply with PLOS ONE's formatting requirements, including file naming conventions.

2. Please note that PLOS ONE has specific guidelines on code sharing for submissions in which author-generated code underpins the findings in the manuscript. In these cases, we expect all author-generated code to be made available without restrictions upon publication of the work.

Response: In accordance with the guidelines of PLOS ONE, we uploaded all author-generated code to the figshare database and confirmed that our code adhered to best practices. The DOI for the code is: 10.6084/m9.figshare.28623698. We ensured that the code was publicly accessible without any restrictions for use and validation by other researchers.

Response: Thank you for pointing this out. We provided the correct funding information in the "Funding" section at the end of the Revised Manuscript with Track Changes file: This research was funded by the National Natural Science Foundation of China (No. 41671514), the Natural Science Foundation of Henan Province (No. 162300410122), and the Surveying and mapping science and technology "double first-class" discipline creation project (No. GCCYJ202409, No. BZXCG202403).

4. We note that your Data Availability Statement is currently as follows: All relevant data are within the manuscript and its Supporting Information files. Please confirm at this time whether or not your submission contains all raw data required to replicate the results of your study. Authors must share the “minimal data set” for their submission.

Response: Thank you for pointing this out. We uploaded the minimal data set used in our experiments to the figshare database to support the replication of our study's results. The DOI for the data set was: 10.6084/m9.figshare.28623698.

5- We require you to either present written permission from the copyright holder to publish these figures specifically under the CC BY 4.0 license, or remove the figures from your submission: You may seek permission from the original copyright holder of Figure 6 to publish the content specifically under the CC BY 4.0 license.

Response: Thanks. The map images in Figure 6 were based on high-resolution imagery from China’s Gaofen satellites, which had been purchased by our research team. We confirmed that the copyright for these images allowed for their public use and dissemination, including publication under the CC BY 4.0 license for your journal. We had also signed the content-permission form and uploaded it as an attachment to ensure compliance with the CC BY 4.0 license.

6- We note that Figures 7, 8, 9, 10, 11, 12, 13 and 16 in your submission contain copyrighted images. All PLOS content is published under the Creative Commons Attribution License (CC BY 4.0), which means that the manuscript, images, and Supporting Information files will be freely available online, and any third party is permitted to access, download, copy, distribute, and use these materials in any way, even commercially, with proper attribution. We require you to either present written permission from the copyright holder to publish these figures specifically under the CC BY 4.0 license, or remove the figures from your submission:

Response: The copyright status of the map images in Figures 7–13 and 16 was the same as Figure 6, and we followed the same approach.

7- Please review your reference list to ensure that it is complete and correct. If you have cited papers that have been retracted, please include the rationale for doing so in the manuscript text, or remove these references and replace them with relevant current references. Any changes to the reference list should be mentioned in the rebuttal letter that accompanies your revised manuscript. If you need to cite a retracted article, indicate the article’s retracted status in the References list and also include a citation and full reference for the retraction notice.

Response: We carefully checked the reference list and confirmed that no retracted papers were cited in our manuscript. In addition, we revised the reference format to comply with PLOS ONE's journal standards and ensured that the reference list was complete and accurate. In response to Reviewer 2's third comment, we added reference [19] in line 97 of the introduction. Additionally, we removed the original references [26], [27], [28], and [29], and added new references [26] and [27] specifically for the SAM index and mSSIM index.

Responses to the reviewers:

Responses to the Reviewer’s Comments

Reviewer 1

1. Page 22, Datasets. It would be very useful if this algorithm is applied to a large area, for example an area of 30 km by 30 km. Can you do this experiment?

Response: Thank you for your suggestion. In our original approach, we divided a 7.3 km × 7.3 km remote sensing image into 128 × 128 pixel (meter) patches and trained the CGAN model on a laptop with an 8GB GPU, where the maximum memory usage reached 5.9GB. Training for 50 epochs and making predictions took a total of 24 minutes. We then conducted experiments on a high-performance computer with a 16GB GPU, dividing a 30 km × 30 km remote sensing image into 128 × 128 pixel (meter) patches. The maximum memory usage reached 14.9GB, and the experiment was successfully executed, yielding promising results. The total time for training 50 epochs and prediction on this large-area image was 19 minutes. These findings demonstrate that our method can effectively process large-area remote sensing images.

2. Page 32, Discussion. Discuss the limitations of your method and draw some future directions.

Response: We moved the discussion of the limitations and future directions from the end of the Conclusion section to Section 4.4 of the Discussion and expanded this section accordingly. The revised content can be found in Section 4.4.

Reviewer2

1. The abstract should include a reflection of the specific indicators and numerical values of the experimental results.

Response: Thanks. We revised the abstract to include specific experimental results as suggested. The updated version now presents quantitative indicators such as RMSE, SAM, mSSIM, and CC. Please refer to the abstract for details.

2. At the end of the abstract, a sentence should be provided stating the significance of this study for the field.

Response: Thank you for your suggestion. We revised the final part of the abstract accordingly. The updated version now concludes with the following statement: " The effectiveness of the proposed method will improve the quality of cloud removal in optical remote sensing images."

3. The sentence 'It can be confidently stated that' in line 97 of the introduction is not suitable for use in the paper. Please cite appropriate references as support.

Response: We revised the original sentence to "Regardless of how complex the terrain conditions of a region were, or how high the temporal resolution requirements for the studied objects in optical remote sensing images might have been, it was always possible to find cloud-free optical remote sensing images for areas that were cloud-covered." Additionally, we supplemented reference [19] to support the statement.

4. Before introducing the contribution of this article, provide a more clear transition.

Response: Thank you for your suggestion. We revised the content as follows: " However, existing methods have yet to effectively integrate SAR image features with neighboring optical image features for cloud removal in optical remote sensing images, thus limiting the enhancement of cloud removal performance. To tackle this issue, this study proposes a novel cloud removal method that integrates SAR and neighboring optical remote sensing image features."

5. Restructure Key Contributions: It may be worth condensing these points into clear contributions and then elaborate on them further in the document.

Response: Thanks. We made the corresponding revisions as suggested, and the updated key contributions can be found in the last section of the Introduction.

6. It is necessary to improve the resolution of Figure 6. In addition, the clarity of the experimental results figures in the article is poor. Please improve the figures quality appropriately�you can convert the image into a vector image in SVG format.

Response: Thanks. We revised all the images by converting them to TIF format with an improved resolution of 330 dpi and uploaded them as attachments.

7. It is necessary to discuss the unpredictable results or patterns encountered in this study, as well as the limitations of this research.

Response: Thanks. We moved the discussion of the limitations and future directions from the end of the Conclusion section to Section 4.4 of the Discussion and expanded this section accordingly. The revised content can be found in Section 4.4.

---

## [Decision Letter · Decision Letter 1]

6 Jun 2025

Dear Dr. Cheng,

The manuscript must be corrected in all points indicated by the reviewers, such as:

1) L53 to L55: It is proven that… here you need a reference containing this proof

2) L54: Can you elaborate a bit (here or in any suitable part) about the types of cloud the proposed methodology addresses? Cloud thickness, altitude etc?

3) L56: Give an example of what kind of auxiliary data are needed

4) L64: One general comment, remote sensing appears many times. Use abbreviation RS instead

5) L69: Some researchers propose… this is not a sentence appropriate for a research article. Be specific on who proposes what with citations.

6) L123: The text inside the figure can not be read. Make it larger (same font size as the text or at least same as the caption)

7) L166: Figure 2 is way down the road. Move it closer to the cross-reference

8) [L139–156] : a - Clarify whether patches are from the same date or synthesized (e.g., “cloudy patches from t1 and cloud-free from t0 or t2”). b - Add more information on cloud types in GF1-WHU dataset (e.g., vertical extent, stratification), not just "various conditions". c - Lacks detail on how image pairs are selected or validated. Describe what augmentations were used: flipping? rotation? brightness?

9) [L160]: Add a sentence comparing ViT and CNNs specifically for SAR

10) [L224–226]: Be explicit about how reconstructed and non-reconstructed regions are defined in practice. Add a figure or schematic if possible.

11) [L228–229]: Clarify how spectral mapping is learned: supervised training? Patch loss metrics?

12) [L233–261]: The CGAN structure is described clearly, well done but needs justification of design choices (e.g., why no batch norm?).

13) [L310–337]: a - Incomplete description of preprocessing steps. b -  Lacks justification for scene selection (e.g., why Area A vs B?). c - No mention of data licensing, public access, or reproducibility.

We look forward to receiving your revised manuscript.

Kind regards,

Claudionor Ribeiro da Silva

Academic Editor

PLOS ONE

Journal Requirements:

Reviewers' comments:

Reviewer's Responses to Questions

**Comments to the Author**

Reviewer #2: All comments have been addressed

Reviewer #3: All comments have been addressed

2. Is the manuscript technically sound, and do the data support the conclusions?

Reviewer #2: Yes

Reviewer #3: Yes

3. Has the statistical analysis been performed appropriately and rigorously?

Reviewer #2: Yes

Reviewer #3: Yes

4. Have the authors made all data underlying the findings in their manuscript fully available?

Reviewer #2: Yes

Reviewer #3: Yes

5. Is the manuscript presented in an intelligible fashion and written in standard English?

Reviewer #2: Yes

Reviewer #3: Yes

Reviewer #2: (No Response)

Reviewer #3: Dear authors,

I carefully read your work and here you can find some suggestions that to my opinion could enhance your manuscript.

L53 to L55: It is proven that… here you need a reference containing this proof

L54: Can you elaborate a bit (here or in any suitable part) about the types of cloud the proposed methodology addresses? Cloud thickness, altitude etc?

L56: Give an example of what kind of auxiliary data are needed

L64: One general comment, remote sensing appears many times. Use abbreviation RS instead

L69: Some researchers propose… this is not a sentence appropriate for a research article. Be specific on who proposes what with citations.

L71 to L96 nice review

L123: The text inside the figure can not be read. Make it larger (same font size as the text or at least same as the caption)

L162 Vit is already defined

L166: Figure 2 is way down the road. Move it closer to the cross-reference

[L139–156] :

• Clarify whether patches are from the same date or synthesized (e.g., “cloudy patches from t1 and cloud-free from t0 or t2”).

• Add more information on cloud types in GF1-WHU dataset (e.g., vertical extent, stratification), not just "various conditions".

• Lacks detail on how image pairs are selected or validated. Describe what augmentations were used: flipping? rotation? brightness?

[L160]: Add a sentence comparing ViT and CNNs specifically for SAR

[L224–226]: Be explicit about how reconstructed and non-reconstructed regions are defined in practice. Add a figure or schematic if possible.

[L228–229]: Clarify how spectral mapping is learned: supervised training? Patch loss metrics?

[L233–261]: The CGAN structure is described clearly, well done but needs justification of design choices (e.g., why no batch norm?).

[L310–337]:

• Incomplete description of preprocessing steps.

• Lacks justification for scene selection (e.g., why Area A vs B?).

• No mention of data licensing, public access, or reproducibility.

In general I believe that it is a well structured methodology and well written manuscript that with some easy to address changes it would be suitable for publication.

Thank you for providing me the chance to revise your work and good luck.

Best,

Kostas

**Do you want your identity to be public for this peer review?** For information about this choice, including consent withdrawal, please see our Privacy Policy

Reviewer #2: No

Reviewer #3: **Yes: ** Konstantinos Vasileios Karyotis

---

## [Author Response · Author response to Decision Letter 2]

19 Jun 2025

Responses to the Reviewer’s Comments

Reviewer3

We sincerely apologize for the discrepancy between the line numbers mentioned in the reviewers’ comments and those in our manuscript. To help accurately locate the corresponding revisions, we have added the updated line numbers from the resubmitted manuscript in each response.

1. L53 to L55: It is proven that… here you need a reference containing this proof.

Response: Thank you for your comment. We have revised the sentence in Lines 62–63 and added appropriate supporting references [6], [7], and [8] to provide evidence for the statement.

2. L54: Can you elaborate a bit (here or in any suitable part) about the types of cloud the proposed methodology.

Response: Thank you for your suggestion. We have elaborated on the types of clouds that the proposed methodology can handle in Lines 106–111 of the revised manuscript, specifically within the sentence beginning with “Compared with traditional methods that...”. We clarified that the method is applicable to both thin clouds and thick clouds.

3. L56: Give an example of what kind of auxiliary data are needed.

Response: Thank you for your comment. The types of auxiliary data required—mainly cloud-free optical images from other time phases or data from other sensor types—have already been clarified in the later sections of the manuscript where the introduction and methods are described.

4. L64: One general comment, remote sensing appears many times. Use abbreviation RS instead.

Response: Thank you for the helpful suggestion. Except for the abstract, we have replaced all occurrences of “remote sensing” with the abbreviation “RS” throughout the manuscript.

5. L69: Some researchers propose… this is not a sentence appropriate for a research article. Be specific on who proposes what with citations.

Response: Thank you for your valuable suggestion. We have revised the sentence in Lines 81–83 to specify the researchers and their proposed methods with appropriate citations.

6. L71 to L96 nice review.

Response: We appreciate your positive feedback.

7. L123: The text inside the figure can not be read. Make it larger (same font size as the text or at least same as the caption).

Response: Thank you for pointing this out. We have adjusted the font size within the figure to ensure that all text is clearly legible.

8. L162 Vit is already defined.

Response: We have revised the text to ensure that the abbreviation “Vit” is used consistently after its initial definition.

9. [L139–156]:

(a) Clarify whether patches are from the same date or synthesized (e.g., “cloudy patches from t1 and cloud-free from t0 or t2”).

(b) Add more information on cloud types in GF1-WHU dataset (e.g., vertical extent, stratification), not just "various conditions".

(c) Lacks detail on how image pairs are selected or validated. Describe what augmentations were used: flipping? rotation? brightness?

Response: Thank you for your valuable suggestions. We have carefully addressed each point as follows:

(a) As clarified in Lines 167–169 of the revised manuscript, all image patches referred to here are from the same date, ensuring temporal consistency..

(b) In Lines 176–181, we have added detailed descriptions of the cloud types included in the GF1-WHU dataset, including their vertical extent and stratification characteristics, to provide a clearer understanding of the dataset’s diversity in cloud conditions.

(c) We have provided additional details on the selection and validation of image pairs in Lines 167–170. Furthermore, the data augmentation techniques used in our study—such as flipping, rotation, and brightness adjustment—are now explicitly described in Lines 184–186.

10. [L160]: Add a sentence comparing ViT and CNNs specifically for SAR.

Response: To address this point, we have added a detailed comparison between ViT and CNNs in the context of SAR image global feature extraction in Lines 220–232. This addition helps clarify the rationale for using ViT by highlighting its advantages in handling SAR data.

11. [L224–226]: Be explicit about how reconstructed and non-reconstructed regions are defined in practice. Add a figure or schematic if possible.

Response: Thank you for the suggestion. We have explicitly defined the reconstructed and non-reconstructed regions in practice in Lines 266–270 of the revised manuscript. In addition, we have included a visual illustration in Figure 1, where the red box corresponds to the reconstructed region and the yellow box represents the non-reconstructed region.

12. [L228–229]: Clarify how spectral mapping is learned: supervised training? Patch loss metrics?

Response: Thank you for your comment. We have clarified the learning process of spectral mapping in Lines 274–279 of the revised manuscript. Specifically, the spectral mapping is learned through supervised training using the L1 loss function.

13. [L233–261]: The CGAN structure is described clearly, well done but needs justification of design choices (e.g., why no batch norm?).

Response: Thank you for your positive feedback. We have added a justification of the design choices for the CGAN generator, including the rationale for omitting batch normalization, in Lines 296–303 of the revised manuscript..

14. [L310–337]:

(a) Incomplete description of preprocessing steps.

(b) Lacks justification for scene selection (e.g., why Area A vs B?).

(c) No mention of data licensing, public access, or reproducibility..

Response: Thank you for your valuable suggestions. We have addressed each point as follows:

(a) A complete description of the data preprocessing steps has been added in Lines 397–402 of the revised manuscript.

(b) We have provided detailed justification for the scene selection in Lines 377–385.

(c) Information regarding data licensing, public accessibility, and reproducibility has been included in Lines 402–404.

---

## [Decision Letter · Decision Letter 2]

8 Sep 2025

A novel cloud removal method by fusing features from SAR and neighboring optical remote sensing images

PONE-D-24-59137R2

Dear Dr. Cheng,

We’re pleased to inform you that your manuscript has been judged scientifically suitable for publication and will be formally accepted for publication once it meets all outstanding technical requirements.

Kind regards,

Claudionor Ribeiro da Silva

Academic Editor

PLOS ONE

Additional Editor Comments (optional):

Reviewer #2:

Reviewer #4:

Reviewers' comments:

Reviewer's Responses to Questions

**Comments to the Author**

Reviewer #2: All comments have been addressed

Reviewer #4: All comments have been addressed

2. Is the manuscript technically sound, and do the data support the conclusions?

Reviewer #2: Yes

Reviewer #4: Yes

3. Has the statistical analysis been performed appropriately and rigorously?

Reviewer #2: Yes

Reviewer #4: Yes

4. Have the authors made all data underlying the findings in their manuscript fully available?

Reviewer #2: Yes

Reviewer #4: Yes

5. Is the manuscript presented in an intelligible fashion and written in standard English?

Reviewer #2: Yes

Reviewer #4: Yes

Reviewer #2: (No Response)

Reviewer #4: The previous suggestions have been modified. The manuscript is presented in an intelligible fashion and written in standard English.

It should be accepted in present form.

**Do you want your identity to be public for this peer review?** For information about this choice, including consent withdrawal, please see our Privacy Policy

Reviewer #2: No

Reviewer #4: **Yes: ** Zhao Chen

---

## [Editor Report · Acceptance letter]

PONE-D-24-59137R2

PLOS ONE

Dear Dr. Cheng,

I'm pleased to inform you that your manuscript has been deemed suitable for publication in PLOS ONE. Congratulations! Your manuscript is now being handed over to our production team.

Kind regards,

on behalf of

Dr. Claudionor Ribeiro da Silva

Academic Editor

PLOS ONE